# Earliest long-necked sauropterygian *Lijiangosaurus yongshengensis* and plasticity of vertebral evolution in sauropterygian marine reptiles

Wei Wang [1], Qinghua Shang[1] ✉, Jiansheng Wang[2], Hongke Zi[3] & Chun Li[1]

A long neck is a morphological innovation in vertebrates, particularly iconic in many plesiosaurs, while the function of these long necks in plesiosaurs remains controversial. Here, we report *Lijiangosaurus yongshengensis* gen. et sp. nov. from a previously unknown early Middle Triassic locality in southwestern China. This taxon represents the earliest known sauropterygian evolving an exceptionally long neck with 42 cervical vertebrae, and is identified as a nothosaur rather than the immediate ancestors of plesiosaurs. Our discovery demonstrates that extreme cervical elongation developing more than 30 cervical vertebrae emerged in sauropterygians prior to the rise of plesiosaurs and their pistosaur ancestors. Furthermore, *Lijiangosaurus* possesses a unique type of accessory intervertebral articulation compared with other reptiles, and we attribute this structure to reducing body undulation. This discovery increases the known diversity of accessory intervertebral articulations in reptiles, and underscores the high plasticity of the vertebral column in the early evolution of sauropterygians.

The early Mesozoic marked a pivotal period of faunal revolution following the end-Permian extinction, witnessing the birth of many evolutionarily important animal groups in life history[1]. Sauropterygia arose as a major marine reptilian clade during the Early to Middle Triassic, persisting as key components of the Mesozoic marine ecosystem for approximately 180 million years[2]. The early-diverging sauropterygians include placodonts, pachypleurosaurs, nothosaurs, and basal pistosaurs[3–5]. Plesiosaurs, a symbolic group of extinct reptiles, represent a late-diverging clade from pistosaurs within Sauropterygia. The nothosaurs mentioned in this study are equal to nothosaurians in systematic paleontology, which not only encompass the family Nothosauridae (*Nothosaurus* and *Lariosaurus*), but also other taxa within Nothosauria[4]. Despite numerous described species[4,6], nothosaurs display lower diversity at the genus level and in anatomical morphology compared to other sauropterygian subgroups[5,7,8]. The body sizes of nothosaurs are typically larger than pachypleurosaurs but smaller than pistosaurs (including plesiosaurs)[2,3].

Plesiosaurs are usually characterized by their spectacularly long necks[2,9]. Despite the secondary development of short necks in some later-diverging plesiosaurian taxa[10,11], early plesiosaurs and their Triassic ancestral kin, the basal pistosaurs, possess impressive elongate necks[12] when the number of cervical vertebrae over 30 is suggested as a synapomorphy of basal pistosaurs and plesiosaurs[13]. Following this traditional view, we consider only a neck with more than 30 cervical vertebrae as a long or elongate neck in this study. This iconic character in plesiosaurs is distinctive in secondarily marine tetrapods, when other representatives like ichthyosaurs, thalattosuchians, mosasaurs, and cetaceans are all short-necked and more fish-like[14]. Certain early-diverging archosauromorph taxa (e.g., *Dinocephalosuarus*, *Tanystropheus*) convergently achieved general body plans resembling those of long-necked plesiosaurs, while these archosauromorphs evolved fundamentally distinct vertebral morphologies[15]. Compared to other eosauropterygian groups that became extinct during the Late Triassic or even at the end of the Middle Triassic[5], such as pachypleurosaurs and nothosaurs, it is unclear whether the remarkably long neck of pistosaurs was a key evolutionary innovation that enabled their survival through the end-Triassic extinction, even though such a long neck is suggested to be hydrodynamically disadvantageous for their locomotion[16].

Here, we report a Triassic sauropterygian skeleton from a locality in western Yunnan province near the eastern Tibetan Plateau and northern Myanmar. This site differs from the previously documented fossil-rich regions[17] in southwestern China around the boundary between Yunnan and

[1]Institute of Vertebrate Paleontology and Paleoanthropology, Chinese Academy of Sciences, Beijing, China. [2]Yunnan Biantun Cultural Museum of Yongsheng, Lijiang, Yunnan, China. [3]Lijiang Municipal Administration of Culture and Tourism, Lijiang, Yunnan, China. ✉e-mail: shangqinghua@ivpp.ac.cn

Guizhou provinces (Figs. S1 and S2). A taxon is erected, *Lijiangosaurus yongshengensis*, which is unambiguously assigned to nothosaurian but exhibits a conspicuously long neck with more than 40 cervical vertebrae, twice the number observed in most coeval sauropterygians[5,8,10]. Even though neck elongation and body plan in sauropterygians have been extensively studied[5,8,10,11,16,18], this unexpected morphology of *Lijiangosaurus* revises our knowledge of the cervical elongation and the vertebral adaptation to this change in Triassic sauropterygians.

## Results
### Systematic paleontology
Reptilia Linnaeus, 1758
Diapsida Osborn, 1903
Sauropterygia Owen, 1860
Nothosauria Baur, 1889

**Emended definition**. Referred to the phylogenetic hypotheses recovered in this paper, the maximum clade definition of Nothosauria is reformulated as follows: all taxa more closely related to *Brevicaudosaurus jiyangshanensis*, *Germanosaurus schafferi*, and *Nothosaurus* species than *Keichousaurus hui*, *Simosaurus gaillardoti*, or *Corosaurus alcovensis*.

**Composition**. Based on the phylogenetic hypotheses in this study, Nothosauria includes *Brevicaudosaurus jiyangshanensis*, *Germanosaurus schafferi*, *Lijiangosaurus yongshengensis*, *Wangosaurus brevirostris*, and all the species of *Nothosaurus* and *Lariosaurus*.

**Diagnosis**. Nothosauria is a clade distinguished from other eosauropterygians by a combination of the following characters: dorsal exposure of prefrontal reduced, pineal foramen displaced posteriorly, mandibular articulation approximately at the level of the occipital condyle, diastema between premaxillary and maxillary teeth present, and four or more sacral ribs.

### *Lijiangosaurus yongshengensis* gen. et sp. nov
**Life Science Identifier (LSID)**. urn:lsid:zoobank.org:act:54B7A70D-DF18-496D-8796-D0EF8A6AECE2

**Etymology**. Both the generic and species names refer to the fossil site of Yongsheng County, Lijiang City, Yunnan Province, China. This currently only known specimen of this genus and species is the first Mesozoic reptile collected from this area.

**Holotype**. YSBB208, preserved on a massive limestone block and deposited at the Yunnan Biantun Cultural Museum of Yongsheng in Chenghai close to Lijiang city, is an incomplete skeleton of a single individual, including the skull, most parts of the vertebral column especially a complete cervical series, most elements of the appendicular bones, all of which are largely in articulation (Figs. 1 and S3).

**Horizon and locality**. Anisian, the early Middle Triassic, Beiya Formation; Banqiao village, Shunzhou Town, Yongsheng County, Lijiang City, Yunnan Province, southwestern China. The Beiya Formation is comparable to the Guanling Formation, and this fossil is coeval with previously known marine reptiles from eastern Yunnan and western Guizhou provinces (e.g., Luoping, Luxi, and Panzhou localities)[17]. However, the fossil studied here is from a site in western Yunnan on the western side of the Khamdian Oldland and expands the distribution of Triassic marine fauna in the South China Block (Figs. S1 and S2).

**Diagnosis**. A medium-to-large-sized nothosaurian distinguished from other nothosaurians by a proportionally small skull with body length over 2.5 m (smaller than a few species of *Nothosaurus* like *N. giganteus* and *N. mirabilis*, but larger than most nothosaurian taxa), a remarkably high count of 42 cervical vertebrae, a dorsal neural spine comparable to the corresponding centrum in height (distinctly shorter than *N. mirabilis*, but longer than all other nothosaurians), entepicondylar foramina lost, accessory intervertebral articulation of infraprezygapophysis and infrapostzygapophysis present in dorsal and anterior caudal vertebrae. It can be further distinguished from non-nothosaurian eosauropterygians (especially pistosauroids) by the absence of interpterygoid vacuity, a ventral flange along the ventromedial edge of the quadrate ramus of the pterygoid, well-developed and extending further posteriorly up to the quadrate, an obturator foramen opening in the adult, and other synapomorphies of nothosaurians.

**Description and comparison**
Although only parts of the dermal palate and the occiput of the skull in YSBB208 are recognizable (Figs. 1 and S4), its skull configuration can be confidently identified to correspond more closely to *Nothosaurus* species[19] than other Triassic marine reptiles[5]. The paired pterygoids form most of the palatal surface of the skull (Fig. S4). They show a continuous suture along the midline of the dermal palate, without any remnant of an interpterygoid fenestra as in pistosaurs (including plesiosaurs)[4,12]. Further posteriorly, the entire basicranium is covered by the pterygoid in ventral view, and the pterygoids meet by a deeply interdigitating suture in their posterior-most part, resembling the species of *Nothosaurus*[19–21] and *Cymatosaurus*[22,23]. A distinctly projecting ventral flange extends along the ventromedial margin of the anterior part of the quadrate ramus of the pterygoid, extending up to the quadrate as in *Nothosaurus*[20] and a species of *Cymatosaurus*[23], which serves the attachment of the deep and superficial layers of the pterygoid muscle. The quadrate suffered significant damage and only its lateral condyle is exposed. From the remaining quadrant, the skull shape and the mandibular joint location can be determined. As in *Nothosaurus*[20] and *Wangosaurus*[24], the mandibular joint lies on a level with the occipital condyle in this specimen, differing from those that are posterior to the level with the occipital in pistosaurs[4,12,25] and most of the other eosauropterygians[4,26,27]. Large, well-defined eustachian foramina are situated on either side of the skull between the basioccipital tuber and pterygoid, resembling those in *Simosaurus* and *Nothosaurus*[20,27,28]. Enlarged fangs and smaller conical teeth are developed on both the upper and lower jaws (see more description of dentition in Supplementary Information and Fig. S4).

The complete cervical and the anterior dorsal series of the vertebral column remain in articulation, preserving a total of 48 vertebrae exposed in ventral or ventrolateral view. We distinguish the cervical vertebrae from the dorsal vertebrae based on articulation facets to the rib heads on the lateral surface of the centra and neural arches[29–33]. The parapophysis for the attachment of the capitulum on the rib is located on the centrum until the 42nd vertebra. On the 43rd vertebra, only a single articulation for the rib is entirely developed on the laterodorsal margin of the centrum abutting the neural arch and displaying transitional morphology. From the 44th vertebra onwards, this entire facet fuses with the diapophysis to be the transverse process located on the neural arch (Figs. S5 and S6). The relatively larger articulation facet on the 44th vertebra likely provides the attachment of a single-headed large rib, which probably distally contacts the sternum and can be interpreted as a dorsal rib[2,3]. Furthermore, the 41st to the 43rd vertebrae are located between the elements of the pectoral girdle, conforming the delimitation between the cervical and dorsal regions (Figs. 1 and S5 and S6). Therefore, the number of cervical vertebrae, which articulate to the bicipital cervical ribs on both centra and neural arches, should be 42.

The spherical anterior articular surface of the atlas in this specimen (Figs. S5 and S6) is similar to *Neusticosaurus pusillus*[34] when other eosauropterygian specimens display this detail, while the atlas centrum in *N. pusillus*[34] is almost as long as the axis centrum, and its posterior part is distinctly constricted laterally. The atlas and axis here are not combined in a complicated manner as in *Bobosaurus forojuliensis* with discernible intercentra of both atlas and axis[35]. However, the atlas and axes of Triassic sauropterygians are rarely observed, and the taxonomic implications of the anatomical morphology of these two elements are uncertain. From the third to the 42nd cervical vertebra,

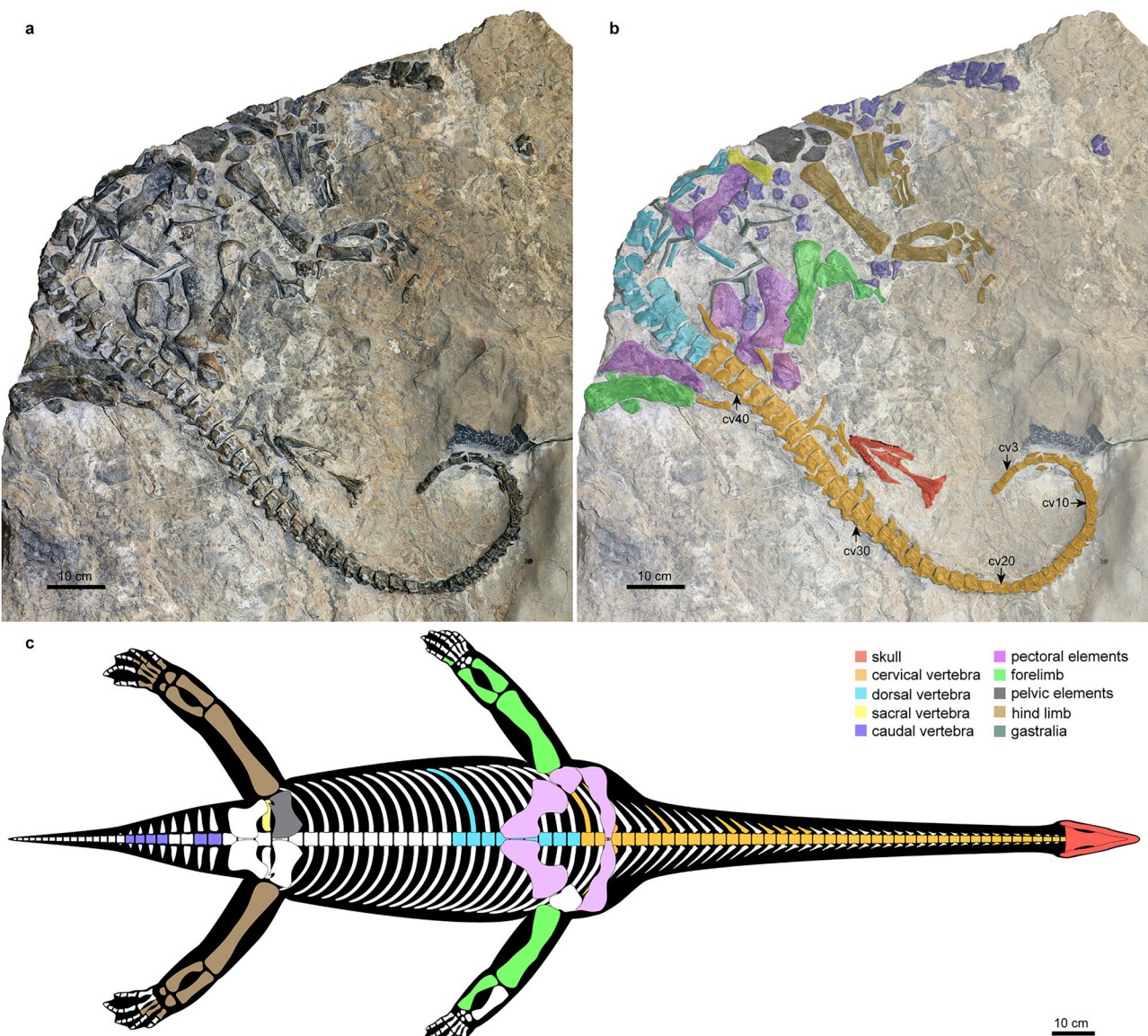

**Fig. 1 | Type specimen (YSBB208) of *Lijiangosaurus yongshengensis* nov. gen. et sp.** Image of the skeleton on a large block of limestone (**a**); interpreted illustration of the skeleton with different osteological parts labeled and marked in multiple colors (**b**); cv is the abbreviation of cervical vertebra; reconstruction of the complete skeleton in strict accordance with the preserved morphology in corresponding colors (**c**); scale bars equal 10 cm.

the length and width of the cervical vertebrae increase slightly and gradually, with the former increasing much less than the latter. Subcentral foramen is absent in any of the cervical centra (Fig. 1; Figs. S5 and S6). Neural arches can be observed on some laterally exposed cervical vertebrae, and their pre- and postzygapophyses are essentially horizontally oriented, but the neural spines cannot be observed due to the preservation.

Only the anterior six dorsal vertebrae and the neural arches of the 7th and the 9th dorsal vertebrae are preserved in articulation (Figs. 1; S5 and S6). The number of dorsal vertebrae is conservative in Triassic nothosaurs and pistosaurs at about 20 to 24[3,4,12,36]. Based on the distance between the 9th dorsal vertebra to the pelvic girdle and the lengths of the preserved dorsal centra, we tentatively consider that there are approximately 20 dorsal vertebrae in this animal. Compared to the cervical vertebrae, the pre- and postzygapophyses of the dorsal vertebrae are much larger and have more vertical inclination with an anteroposterior trend of increasing. On the neural arch, at least clearly of the 4th to the 7th dorsal vertebra, a projection is present below and between the postzygapophysis. Correspondingly, a cavity is supposed on the anterior face of the succeeding vertebra below the prezygapophysis, although this cavity is covered in the dorsal region, but in the caudal region (Figs. S5 and S6). We

identify this projection as infrapostzygapophysis because it is not as proximally located as the hyposphene in *Placodus gigas*[37] (Fig. 2). The infrapostzygapophysis in this reptile developed more laterally on the neural arch and almost vertically under the postzygapophysis, which is unique in all known sauropterygians except the similar structure present in *Simosaurus* (Fig. 2). A small number of disarticulated dorsal rib fragments are scattered across the trunk, and they are unicapital without pachyostosis.

A sacral rib is preserved and is distinct from the ribs belonging to other regions of the vertebral column, and is short, massive, and bar-like (Fig. 1). Its shaft is straight and broad in dorsoventral view. Its distal end is distinctly expanded with a clear articular facet to the pelvic girdle. This morphology of the sacral rib here is reminiscent of *Nothosaurus*[21] and a possible *Cymatosaurus*[38] instead of *Yunguisaurus*[12]. Posterior to the pelvic girdle and the hind limbs, five neural arches of the proximal caudal region are preserved in tight articulation. Zygapophyses developed more horizontally in these caudal vertebrae than in the dorsal vertebrae. Similar to the dorsal vertebrae, a pronounced infrapostzygapophysis below the postzygapophysis and an obvious infraprezygapophysis as a concavity below the prezygapophysis are present in each of these caudal vertebrae (Fig. 2). A disarticulated neural arch,

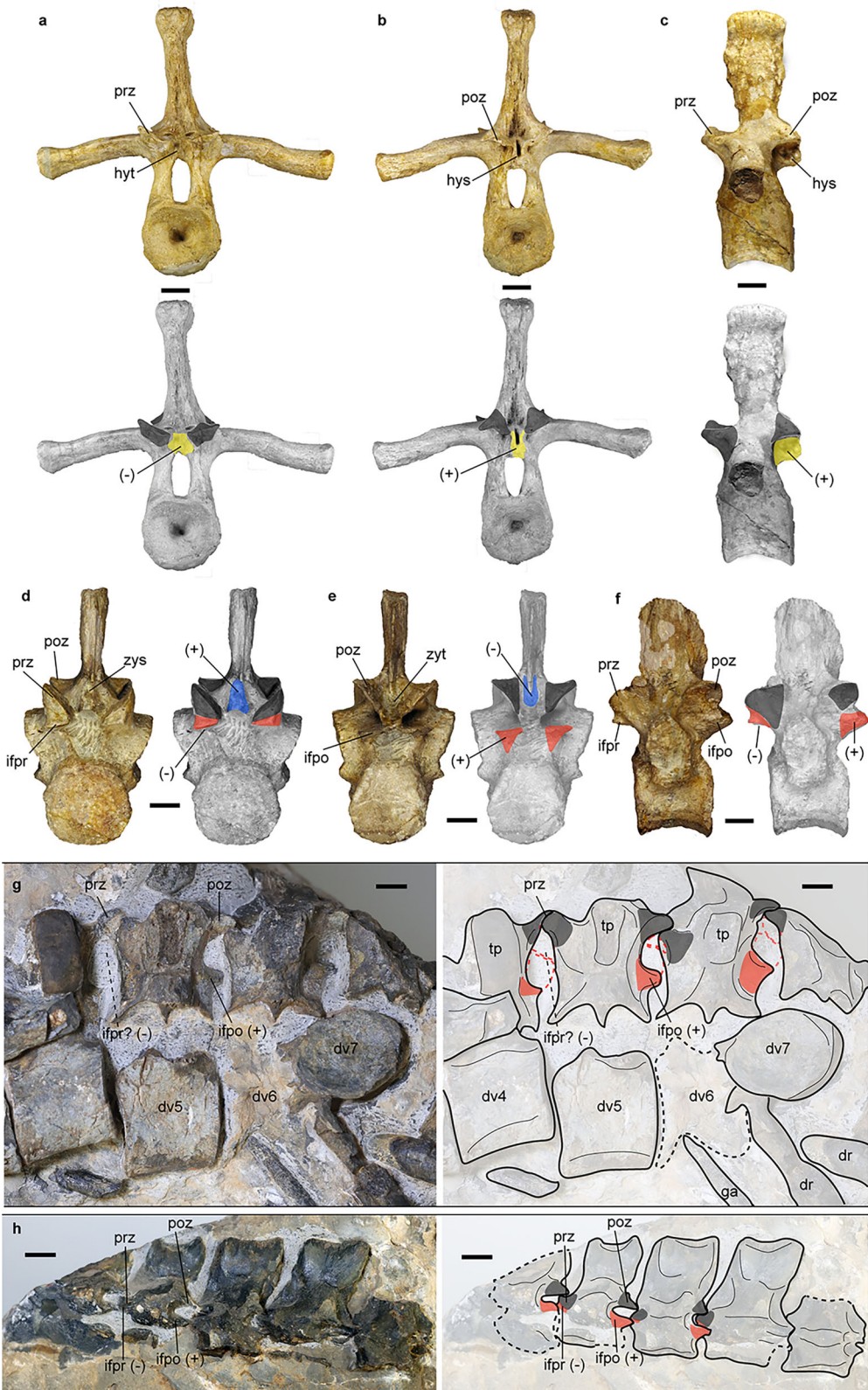

**Fig. 2 | Three types of accessory intervertebral articulation (AIA) in sauropterygian marine reptiles.** Color photographs and faded images of the same vertebrae with different types of AIAs are highlighted in different colors. Taxa and vertebral region: *Placodus gigas* (SMNS 59825), dorsal vertebra in anterior (**a**), posterior (**b**), and left lateral view (**c**); *Simosaurus gaillardoti* (SMNS 14733), dorsal vertebra in anterior (**d**), posterior (**e**), and left lateral view (**f**); *Lijiangosaurus yongshengensis* (YSBB208), dorsal vertebrae in left lateral view (**g**), and proximal caudal vertebrae in left lateral view (**h**). Abbreviations, colors, and symbols: dr, dorsal rib; dv, dorsal vertebra; ga, gastralium, hys, hyposphene (yellow, ( + )); hyt, hypantrum (yellow, (-)); ifpo, infrapostzygapophysis (red, ( + )); ifpr, infraprezygapophysis (red, (-)); poz, postzygapophysis (gray); prz, prezygapophysis (gray); tp, transverse process; zys, zygosphene (blue, ( + )); zyt, zygantrum (blue, (-)); ( + ), convex articulatory surface; (-), concave articulatory surface. Scale bar equals 2 cm in (**g**) and 1 cm in others.

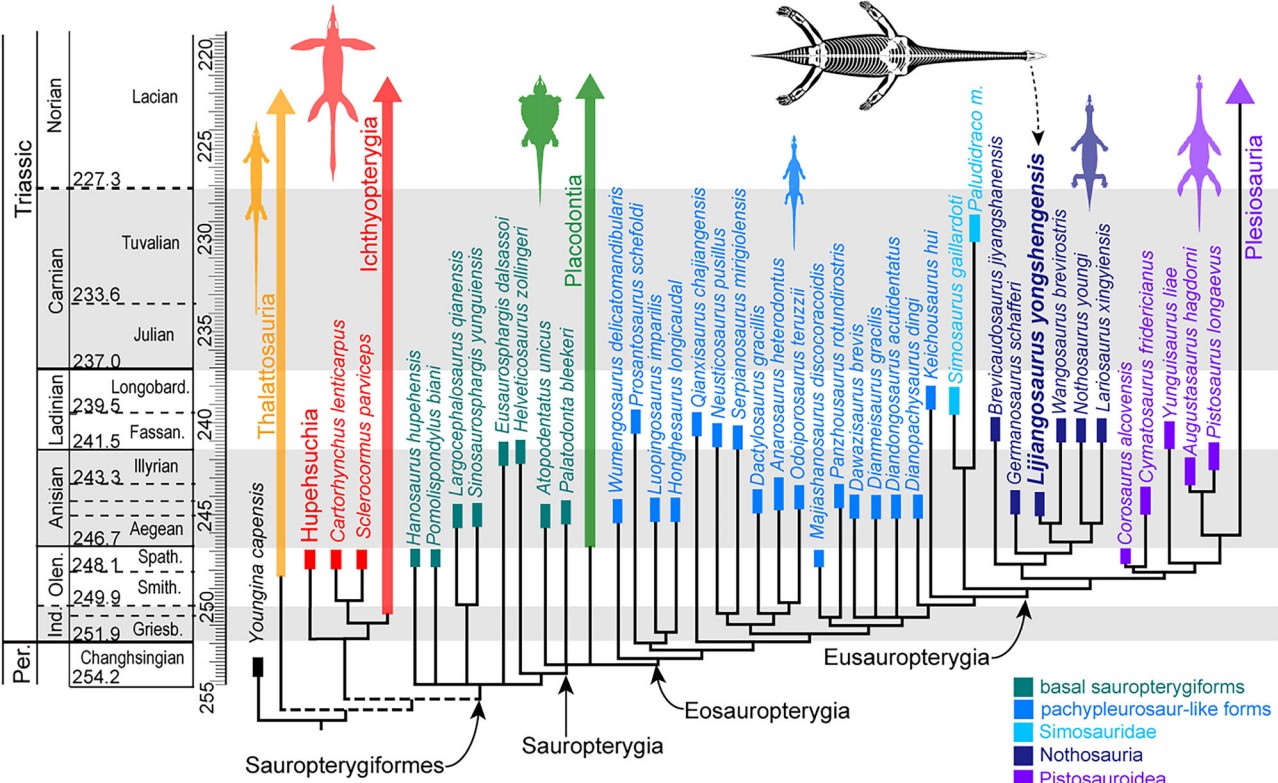

**Fig. 3 | Phylogenetic relationship of *Lijiangosaurus yongshengensis* nov. gen. et sp. with other sauropterygians.** The topology is based on a strict consensus tree from the 16 most parsimonious trees using an updated morphological character dataset from a previous study[5]. *Lijiangosaurus yongshengensis* is the sister group of *Wangosaurus brevirostris*, both belonging to Nothosauria but not Plesiosauria.

probably from a proximal caudal vertebra preserved in lateral view between the pair of coracoids, exhibits a relatively tall neural spine and confirms the presence of an infrapostzygapophysis (Figs. S5 and S6).

The clavicle is a large band-like plate with an irregular outline (Figs. 1; S5 and S6). The relatively complete clavicle, possibly the right one displaced near the left humerus, exhibits a morphology typical in nothosaurs (for example, *Nothosaurus* cf. *N. mirabilis* in Rieppel, 2001: Fig. 8)[19], which is broad and thick possessing a large lateral expansion with a tapering posterolateral process, but it still differs from that in other previously known nothosaurs by developing a slight anteromedial expansion and a distinctly constricted middle portion. The coracoid shows a broad lateral expansion that contacts the glenoid portion of the scapula, a strongly concave anterior margin opposing the less distinctly concave posterior margin, and a medial expansion establishing a broad contact with the other coracoid. This coracoid morphology is typically present in nothosaurs[19] and similar to some small-sized eosauropterygians[39–41] and cf. *Cymatosaurus*[38], but evidently differs from pistosaurs (e.g., *Corosaurus*[36], *Augustasaurus*[32], and *Yunguisaurus*[12]). The coracoid foramen is unknown in this specimen.

The humerus is stout and slightly curved, with both its proximal and distal ends of nearly equal widths. The right humerus is exposed in dorsal view, showing a concavity for the attachment of the teres major and latissimus dorsi muscles[42]. The status of the deltopectoral crest is unknown. Its ectepicondylar groove is open without an anterior notch, while the entepicondylar foramen is absent (Fig. S7). Again, this humerus is more similar to those in nothosaurs[43] than in pistosaurs[12,13] and pachypleurosaur-like forms (see more description on forelimbs in Supplementary Information and Fig. S7).

The lateral portion of the right pubis is preserved (Figs. 1 and S8), displaying weakly concave anterior and posterior margins. The obturator foramen is an open slit at the suture to the ischium near the acetabular margin of the pubis. The pubis seems more rectangular than round without distinct constriction, again reflecting the typical structure in nothosaurs[3,21,44] compared to pistosaurs[12,36] and pachypleurosaur-like eosauropterygians[4,40,41].

The preserved metatarsals develop expanded ends and bar-like shafts as robust as those in certain nothosaurians[6,21,28]. None of the preserved metatarsals and phalanges are shortened or flattened (Figs. 1 and S8), which is different from *Yunguisaurus*[12] and plesiosaurs. Nevertheless, the preservation of manus and pes is too incomplete to fully exclude the development of hyperphalangy in this reptile (see more description on hind limbs in Supplementary Information and Fig. S8).

**Phylogenetic relationships**

To investigate the position of *Lijiangosaurus yongshengensis* nov. gen. et sp. in sauropterygians, we mainly integrate YSBB208 into an updated character matrix focusing on Triassic sauropterygiforms[5], which includes 52 genera of almost all known Triassic sauropterygians and basal sauropterygiforms (Supplementary Data 1), more intensively sampling than many recent matrices[7,18,45]. We additionally test the affiliation of YSBB208 in another independent recent dataset[45] (Supplementary Data 2). In our phylogenetic result (Figs. 3 and S9), *Hanosaurus*, *Eusaurophargis*, and recently reported *Pomolispondylus*[46] are placed at the base of Sauropterygiformes, although they form a polytomy with Saurosphargidae and Sauropterygia, probably due to the limited morphological information of *Pomolispondylus* so far. The clade of Eusauropterygia recovered here is comparable with many previous studies[5,7,8,18,47], while the "pachypleurosaurs" are paraphyletic, involving both eastern and western Tethyan small-to-medium-sized eosauropterygians. Remarkably, an extremely elongate neck developing more than 30 cervical vertebrae, contrary to what was previously suggested[13], is no longer a synapomorphy for pistosaurs or plesiosaurs after our discovery of *Lijiangosaurus*.

Our analytical results support that *Lijiangosaurus* is a member of nothosaurs rather than a basal pistosaur or plesiosaur (Fig. 3; Figs. S9 and S10). In all the most parsimonious trees and the strict consensus tree, *Lijiangosaurus* is consistently the sister group of *Wangosaurus* among nothosaurians (Figs. 3; S9 and S10), and these two taxa collectively become

the sister family to Nothosauridae that comprises all the species of *Nothosaurus* and *Lariosaurus* (Figs. 3 and S9)[4,6].

## Discussion

### Nothosaur affiliation of *Lijiangosaurus yongshengensis*

In the ventral view of the skull, a broad notch diverging from the posteromedial margin of the pterygoids, which is termed the posterior palatine vacuity[48] or the posterior interpterygoid vacuity[3,49], is absent in *Lijiangosaurus*. This vacuity is a synapomorphy shared by plesiosaurs[2,3] and their pistosaur ancestors, such as *Pistosaurus*[4,36], *Augustasaurus*[33], and *Yunguisaurus*[12,50]. Instead, this interpterygoid vacuity is never present in any known nothosaurs, and their posteromedial portions of the paired pterygoids are firmly integrated by interdigitating sutures[4,8,19,27], which is also the case in *Lijiangosaurus*. Additionally, the paired and small nutritive openings present on the lateroventral surface of the vertebral centra, termed subcentral foramen, represent another characteristic feature in certain pistosaurs, such as *Augustasaurus*[32] and *Pistosaurus*, as well as in plesiosaurs[13,36], while it is absent in *Yunguisaurus*, possibly because of its early divergence in pistosaurs[12]. However, the subcentral foramen is absent in the entire vertebral column in *Lijiangosaurus* and other nothosaurians, such as *Nothosaurus* and *Lariosaurus* spp[4,19,51–53]. Furthermore, the morphology of girdle elements, particularly the constricted coracoid and pubis (Fig. 1), in *Lijiangosaurus* resembles those in other nothosaurians, such as some well-known *Nothosaurus*[4,19,21,51] and *Lariosaurus* species[52,53]. Unlike nothosaurians, the coracoids are broader and less constricted in the Triassic pistosaurs[12,31,32,36], when the pectoral girdles are highly specialized in plesiosaurs[2,3]. Moreover, the pubes are rather rounded in two doubtless pistosaur genera, *Yunguisaurus*[12,54] and *Bobosaurus*[35,55,56], distinguishing from these in *Lijiangosaurus* and other nothosaurs. As in nothosaurs but not pistosaurs, the manus and pes in *Lijiangosaurus* are more claw-like than fin-shaped. Although the limbs are incomplete in YSBB208, the long bones in limbs are robust and the phalanges are not flattened, all resembling nothosaurs, whereas these limb elements are usually flattened with hyperphalangy resulting in paddle-like limbs in pistosaurs[12,54], including plesiosaurs. Besides the aforementioned evidence, the occiput configuration, cervical rib articulation, and many other aspects of anatomy collectively demonstrate the nothosaurian affiliation of *Lijiangosaurus*.

Notably, *Lijiangosaurus* shows some similarities with *Cymatosaurus* and *Wangosaurus*. Several well-preserved skulls are identified as different species assigned to *Cymatosaurus*[4,22,23,57], and some show similar palatal construction with *Lijiangosaurus* as well as other nothosaurs, such as the absence of the interpterygoid vacuity, but the diagnosis of *Cymatosaurus* is identified from the dorsal view of the skull, which makes it impossible to further compare with this specimen. Two postcranial skeletons from the Anisian of the early Middle Triassic at the Winterswijk locality are assigned to cf. *Cymatosaurus* or a closely related basal pistosaur[38], solely based on the humerus morphology and histology, while one of them, NMNHL RGM 449487 A, shows some vertebral and girdle elements more similar to those in *Lijiangosaurus* and other nothosaurs[19,21,58] than in pistosaurs[12,36,56]. Hence, the taxonomy of these two skeletons from Winterswijk and the phylogenetic position of *Cymatosaurus* remain ambiguous before more complete material can be collected. *Wangosaurus* is known from a single specimen and initially identified as a pistosaur, but meanwhile, its nothosaur-like characters have been documented[24,59]. The pistosaur affinity of *Wangosaurus* is confirmed by some researchers[53,60,61], whereas a position within nothosaurs is preferred by other recent studies[5,62,63] and this study uses more comprehensive character matrices. Notably, *Lijiangosaurus* is resolved here as the sister group of *Wangosaurus* (Figs. 3, S9 and S10), and these two genera are collectively within Nothosauria and stably close to Nothosauridae, a monophyletic group including *Nothosaurus* and *Lariosaurus* spp. However, the systematics of *Cymatosaurus* and *Wangosaurus* are beyond the scope herein, while more fossil materials and further examination of them are needed to resolve their position.

### Cervical elongation in Triassic sauropterygians

Tetrapods achieve neck elongation through three primary mechanisms: increase cervical vertebral count via modified somitogenesis, elongation of individual cervical somite, or posterior displacement of the cervico-dorsal boundary through altered Hox gene expression[64]. Unlike some other long-necked reptiles[10,15,65], the length of a single cervical vertebra has never remarkably increased in Triassic sauropterygians. Compared with the cervical region, dorsal vertebral counts are relatively constrained in pachypleurosaurs and pachypleurosaurs-like forms from 16 in *Dawazisaurus*[66] to 28 in *Wumengosaurus*[67], and become more conservative in eusauropterygians (nothosaurs and pistosaurs) around 20, except *Brevicaudosaurus* of 14. It is documented that sauropterygian neck elongation preliminarily results from increased cervical vertebral count[10,18], prompting our focus on this aspect (Fig. 4). Despite the large lack of cervical and dorsal ribs in YSBB208, we can readily distinguish the cervical from the dorsal vertebra based on the articulation facets on the vertebra to the ribs, which is in accordance with the condition considering two or one articulation facets on the proximal end of a rib, and the presence of transverse process formed by the conjoined parapophysis and diapophysis, a widely accepted criterion to identify neck and trunk regions[29–31,33]. Based on the featured articulation discerned either on centra or ribs, this diagnosis is consistently used in this study to discuss the cervical vertebral counts in eosauropterygians.

Compared to the ancestral condition of diapsids developing six cervical vertebrae[64] and the basal sauropterygiforms possessing about 15 cervical vertebrae[5,64], the cervical vertebral counts and the neck lengths have more or less increased in all eosauropterygians. In the small-sized (adult body length normally <1 meter) basal eosauropterygian,s including European pachypleurosaurids and Chinese pachypleurosaur-like taxa, the necks are moderately elongated by developing about 20 cervical vertebrae (Fig. 4), though variation exists from 17 in *Dactylosaurus*, 18 in *Qianxisaurus* and *Serpianosaurus*[26,68] to 26 in *Keichousaurus*[39]. Eusauropterygia is a subclade of Eosauropterygia (Fig. 3) represented by middle-to-large-sized eosauropterygian reptiles (adult body length normally >1 meter)[4,5]. In nothosaurians, the neck length was previously known to be comparable with the small-sized eosauropterygians without much specialization, such as about 19 to 24 cervical vertebrae, variable in different *Nothosaurus* and *Lariosaurus* spp[4,19,44,51,53]. However, incorporating *Lijiangosaurus* and reinterpreting *Wangosaurus* within Nothosauria reveals substantially greater neck elongation compared to their ancestral forms[8]. The cervical count of 42 in *Lijiangosaurus* is the highest among nothosaurians (Fig. 4), and even higher than Triassic pistosauroids except *Yunguisaurus*[5] (Fig. 4). The cervical count of 18 in *Corosaurus*, the earliest known and basal-most pistosaur, implies a status inherited and even reduced from its eosauropterygian ancestor. In other pistosaurs, including all plesiosaurs from the Triassic to Cretaceous, a remarkable variation of cervical counts is reported from 13 to 72[3,69], and multiple independent occurrences and reductions of long necks have been identified in Jurassic and Cretaceous plesiosaurs[10,70].

In the aspect of neck length, the significance of the discovery of *Lijiangosaurus* includes: firstly, it updates the highest cervical count to be 42 and the longest neck in nothosaurians and other non-pistosauroid sauropterygians[18]; secondly, within sauropterygian evolution history, *Lijiangosaurus* represents the earliest occurrence of an exceptionally long neck (comprising over 30 cervical vertebrae), predating the emergence of both basal pistosaurs and their plesiosaur descendants[16]; thirdly, given that the function of elongated necks in plesiosaurs and their pistosaur ancestors remains uncertain, the presence of an elongated neck in *Lijiangosaurus*, which is neither a fast swimmer nor a pursuit predator, provides indirectly support for the foraging benefit hypothesis of neck elongation in sauropterygians, facilitating ambush predation and larger feeding ranges[71,72]. *Lijiangosaurus* evolved a strikingly long neck without fin-shaped limbs as in plesiosaurs, and it seems that rather than the long neck[9], the locomotion and pelagic lifestyle are more crucial evolutionary novelties in plesiosaurs. Besides the neck, further plasticity of the vertebral column reflected by *Lijiangosaurus* in sauropterygians is discussed below.

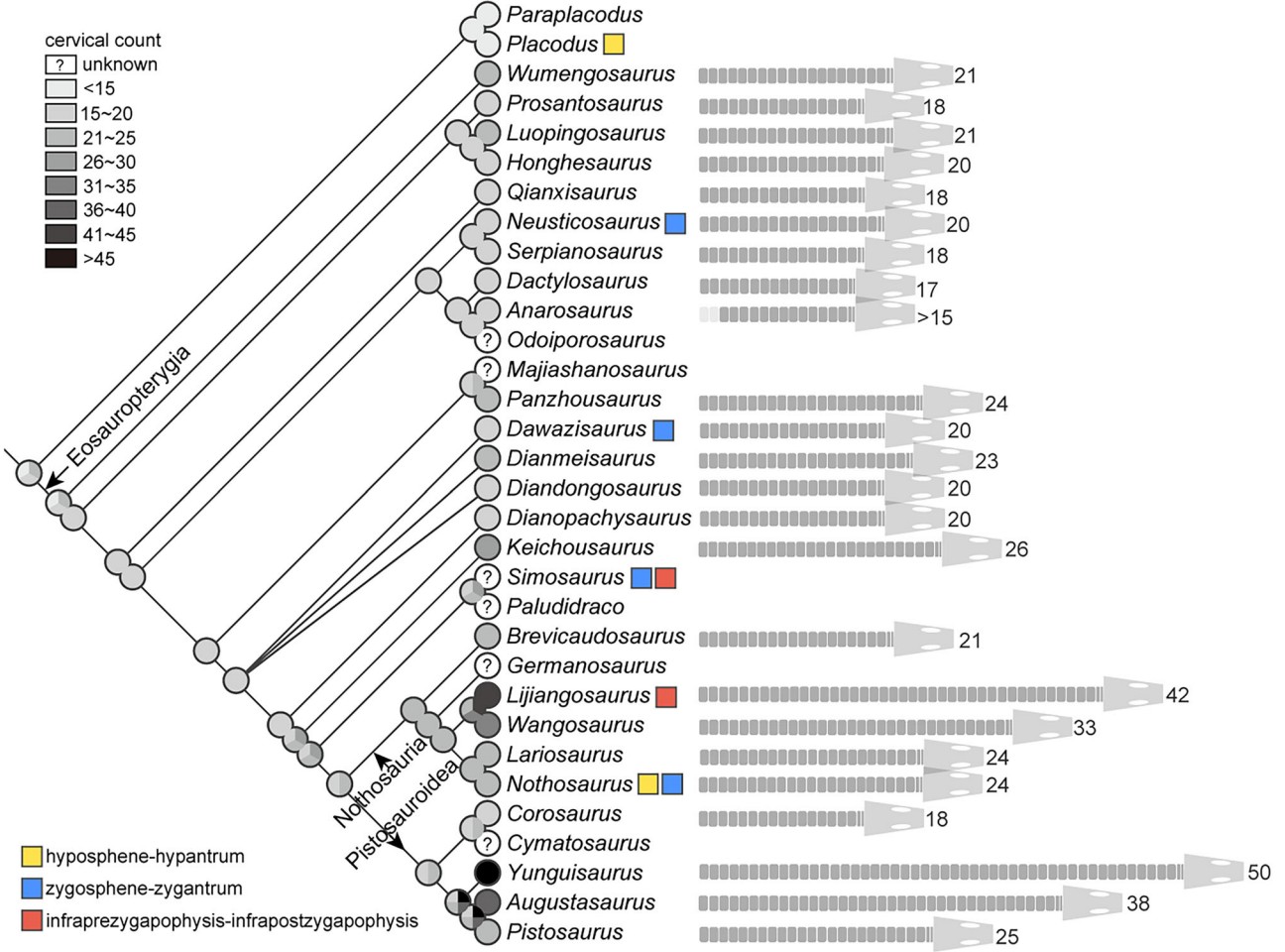

**Fig. 4 | Ancestral state reconstruction on cervical counts and accessory intervertebral articulation types in Triassic eosauropterygian marine reptiles.** Cervical vertebra numbers from less than 15 to more than 45 are reflected by eight grayscales, and the ancestral cervical counts are estimated by percentages of different grayscales, constraining the topology from Fig. 3. Long necks (more than 30 cervical vertebrae) are independently present in nothosaurs and basal pistosaurs. Cervical numbers are illustrated for each taxon when complete necks are preserved. Three types of accessory intervertebral articulations are known in multiple lineages of the Triassic sauropterygian taxa, and the symbol colors follow Fig. 2.

## Plasticity of vertebral articulation in reptiles

One of the diagnostic characters of *Lijiangosaurus yongshengensis* is the accessory intervertebral articulation (AIA) present in the dorsal and the anterior caudal vertebrae, and this structure is innovative and unique in nothosaurs and even in reptiles. In fact, in addition to zygapophyseal articulations, AIA is widely and independently evolved in various reptilian lineages[3]. For better comparisons, we summarize AIA into three categories according to morphologies and locations (Fig. 2). The first kind is hyposphene-hypantrum articulation (HH), when the hyposphene is a vertical bony lamina between the postzygapophyses and the hypantrum is a notch between the prezygapophyses, and both are below the level of the zygapophyseal articular surface[73]. HH is developed in some cotylosaurs[3] and archosaurs, such as certain pseudosuchians and dinosaurs[73]. In sauropterygians, HH is found in basal placodonts (Rieppel, 1995) (Figs. 2 and 4) and some isolated vertebrae possibly belonging to *Nothosaurus* or relevant eusauropterygians[74]. The second type of AIA is zygosphene-zygantrum articulation (ZZ). In contrast to HH, the zygosphene is a convexity between the prezygapophyses, and the zygantrum is a concavity between the postzygapophyses, and both are above the level of the zygapophyseal articulation (Fig. 2). The most prominent and typical ZZ is present in snakes[75], and a similar but smaller AIA is suggested as ZZ in some other lizards and mosasaurs[3]. Previous researchers described ZZ in the cervical vertebrae of a pachypleurosaur-like eosauropterygian, *Dawazisaurus*[66], the dorsal vertebra of a pachypleurosaur, *Neusticosaurus edwardsi*[76], the cervical and dorsal

vertebrae in *Nothosaurus mirabilis*[77] and another vertebra assigned to *Nothosaurus*[63], and isolated vertebrae of *Simosaurus*[78] (Fig. 2), though these are in the same position between and above zygapophyseal processes but not identical in morphology with those in snakes. Notably, besides ZZ, *Simosaurus*[78] shows another AIA termed infraprezygapophysis-infrapostzygapophysis articulation (II), a third type of intervertebral articulation (Figs. 2 and 4). II in *Simosaurus* is almost the same as in *Lijiangosaurus*, when the infraprezygapophysis is a ventral-facing surface immediately under the prezygapophysis, and the infrapostzygapophysis is an independent projection below the postzygapophysis to dorsally articulate with the infraprezygapophysis. Further different from HH, these II laterally extend as wide as the zygapophyseal expansions. On the holotype of *Lijiangosaurus yongshengensis*, these posteriorly protruding AIAs on the dorsal and caudal vertebrae can be identified as II instead of the HH type, although they are only exposed in lateral view. AIA is not identified in many eosauropterygian taxa (Fig. 4), possibly because of the incomplete exposure of vertebrae, and whether there is AIA is uncertain.

The three currently recognized accessory intervertebral articulation (AIA) types appear to have evolved independently in multiple basal sauropterygian lineages (Fig. 4), suggesting functional rather than homologous significance. In other reptile groups, HH is developed in the trunk region correlated with large body size in terrestrial archosaurs, including enormous pseudosuchians and large-bodied sauropod and theropod dinosaurs[73]. Different from these archosaurs with AIA, the aquatic environment for

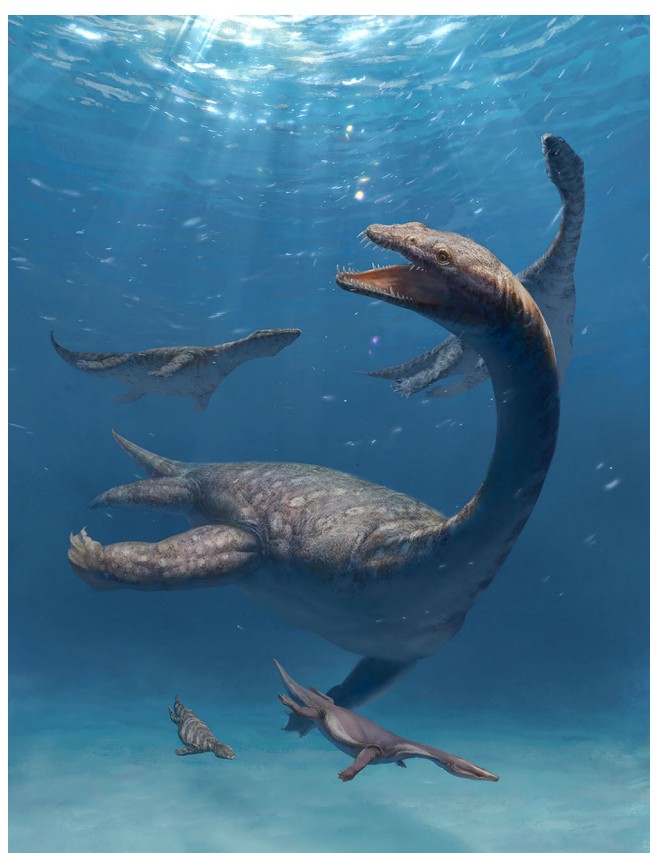

**Fig. 5 | Reconstruction of nothosaurs about 240 million years ago revealing a hidden diversity from southwestern China.** Taxa include *Lijiangosaurus yong-shengensis* (central), *Nothosaurus yangjuanensis* (upper left), *Nothosaurus luo-pingensis* (upper right), *Brevicaudosaurus jiyangshanensis* (lower left), and *Lariosaurus hongguoensis* (lower right). All these nothosaurian taxa lived in the Middle Triassic in Yunnan and Guizhou provinces (artwork by Kelai Li).

sauropterygians alleviates gravitational constraints when their body sizes increase. Moreover, AIA is present in some small-sized eosauropterygians (e.g., *Neusticosaurus edwardsi* and *Dawazisaurus*), and thus the AIA in sauropterygians seems less likely to be a modification or adaptation for gigantism. The function of the ZZ complex in snakes is to prevent axial torsion in the body[76], and this advantage seems plausible in these eosaur-opterygians with AIA[77] because multidirectional movements may more easily occur underwater, caused by varied forces[14]. Besides HH and ZZ articulation in other eosauropterygians, *Lijiangosaurus* presents a novel case with its distinctive II, which we associate with its exceptional neck elonga-tion. The long-necked plesiosaurs evolved rigid trunks by developing broad girdle bones and densely packed gastralia[9,13] to restrict undulation, and this body plan can possibly be beneficial to control the long necks. As a non-pistosauroid eosauropterygian taxon, *Lijiangosaurus* lacks the aforemen-tioned configurations comparable with plesiosaurs or basal pistosaurs for trunk strengthening, while II is evolved and serviceable to reinforce inter-vertebral bracing and reduce rolling in its dorsal and proximal caudal regions. Although vertebral structure is under strong evolutionary con-straint without much modification in reptiles and even all other vertebrates[3], diverse AIAs in eosauropterygians and other reptiles shed light on the plasticity of vertebral morphology.

For a long time, nothosaurs consistently included three European genera, *Germanosaurus*, *Nothosaurus*, and *Lariosaurus*[4,41,60,61,67,79,80], and they are mostly represented by cranial fossil materials that are mainly varied in skull sizes and supratemporal fenestra elongation[4,59]. Supplied with some complete skeletons[19,44,51,53] and more postcranial fossil remains[43,58,77], notho-saurs were considered to develop conservative body shapes[2,4,81] at least

compared to plesiosaurs[10]. Nevertheless, as highlighted by *Brevicaudosaurus*[62] with a stout trunk and long-necked *Lijiangosaurus* in this study, these latest discoveries of nothosaurians from the early Middle Triassic in southern China (Fig. 5) continuously update our knowledge of the early evolution of sauropterygians. These recently reported findings reveal a higher disparity in body plans than previously inferred, which is reflected by the cranial and dental variations[47], the diverse AIA[77] (Fig. 4), the pachyostosis[62] even more prominent than that in pachypleurosaurs, and the elongate necks apparently similar to later plesiosaurs. The highly specialized body shapes of these nothosaurs reflect the evolutionary radiation of sauropterygians during the marine bio-recovery following the end-Permian extinction.

## Methods

### Fossil material and geological context

The type specimen of *Lijiangosaurus yongshengensis* (YSBB208) was ori-ginally found and collected by the local villagers, and then donated to the local museum, Yunnan Biantun Cultural Museum of Yongsheng (YSBB), where it is permanently stored and displayed in the exhibition. The villager, who originally collected the fossil, was not involved in this study, and the specimen was donated to the museum about ten years prior to and independently from our study. This specimen can be observed with the requisite permission from this museum. The block was weathered, and the fossil bones were naturally exposed without artificial treatment until the professional technicians from the Institute of Vertebrate Paleontology and Paleoanthropology (IVPP) of the Chinese Academy of Sciences further prepared it under the supervision of the researchers in this study. Other main fossil material for comparisons in the text and figures is deposited at the collections at the IVPP, listed in Wang et al.[5] and the State Museum of Natural History Stuttgart, Germany (SMNS) (*Placodus gigas* SMNS 59825, *Simosaurus gaillardoti* SMNS 14733).

The matrix of YSBB208 is gray limestone with a layer thickness of more than 1 meter, and this lithology can be readily assigned to the lower part of the Beiya Formation, because this formation overlays the Lamei Formation of purple-red gritstone, and is covered by the upper part of the Beiya For-mation of dolomite, mainly exposed on the western slope of the hill. We discovered several fragments of reptilian vertebrae and some bivalves at that locality, when the latter can be assigned to *Myophoria* (*Costatoria*) *goldfussi mansuyi*-*Eumorphotis* (*Asoella*) *illyrica* fauna and the age of the Beiya Formation is Anisian, early Middle Triassic[82]. The assemblage of the same bivalve species is found in the Yantang Formation[83], which is equivalent to the Guanling Formation[17,84], to further confirm the age of this skeleton to be Anisian (Supplementary Information).

### Phylogenetic analysis and ancestral state reconstruction

The first phylogenetic analysis in this study was mainly based on the character matrix (Supplementary Data 1) constructed by Wang et al.[5], which is considered one of the most comprehensive datasets for elucidating the relationships among sauropterygians. In addition to *Lijiangosaurus yong-shengensis* nov. gen. et sp., six recently reported taxa were incorporated into the original matrix, including *Brevicaudosaurus jiyangshanensis* (IVPP V18625 and V26010)[62], *Honghesaurus longicaudalis* (IVPP V30380)[85], *Luopingosaurus imparilis* (IVPP V19049)[7] that we personally observed, and *Pomolispondylus biani* (WGSC V1701)[46], *Panzhousaurus rotundirostris* (GMPKU-P-1059 and GMPKU-P-3241)[60,61], and *Prosantosaurus scheffoldi* (PIMUZ A/III 1274)[86], with information from detailed descriptions and images in the literature. The updated matrix (Supplementary Data 1) set the early diapsid *Youngina capensis* as the outgroup. To focus on the position of newly added taxa and the interrelationships of eosauropterygians, the taxa of three stem-group turtles and five hupehsuchians were pruned. Parsimony analyses were performed in TNT 1.5[87], treating all characters as equally weighted and unordered. The heuristic search in the traditional search involved 100000 replications of Wagner trees with one random seed, saving 1000 trees in each replication. Tree bisection and reconnection (TBR) was employed in the new technology search with default options[88]. Our analysis resulted in a strict consensus tree (Fig. 3 and Fig. S9) from the 16 most-parsimonious trees at 784 steps (consistency index=0.296, retention

index = 0.707). The second phylogenetic analysis was conducted using the matrix updated from Hu et al.[45], which had the data independent from Wang et al.[5] We additionally scored *Wangosaurus*, *Brevicaudosaurus*, and *Lijiangosaurus* to the original dataset (Supplementary Data 2), and other settings were the same as mentioned above. The strict consensus tree from the second analysis (Fig. S10) was from 3 most-parsimonious tress at 868 steps (consistency index=0.296, retention index=0.604), which confirms the sister-group relationship between *Lijiangosaurus* and *Wangosaurus* both within nothosaurs. The maximum parsimonious ancestral states of the cervical vertebra numbers in eosauropterygians (Fig. 4) were reconstructed in Mesquite 3.81[89] using the constrained topology of the cladogram from the first analysis, in which the cervical counts from less than 15 to more than 45 were divided into eight levels.

## Statistics and reproducibility

The first character matrix[90] comprised 181 characters and 68 taxa (Supplementary Data 1), and the second character matrix[90] involved 203 characters and 46 taxa (Supplementary Data 2). The scores can be confirmed from the morphology of each taxon, and the results can be reproduced following the steps mentioned above.

## Nomenclatural acts

This published work and the nomenclatural acts it contains have been registered in ZooBank, the proposed online registration system for the International Code of Zoological Nomenclature (ICZN). The ZooBank LSID (Life Science Identifiers) can be resolved and the associated information viewed through any standard web browser by appending the LSID to the prefix "http://zoobank.org/". The LSID for this publication is: urn:lsid:zoobank.org:act:54B7A70D-DF18-496D-8796-D0EF8A6AECE2.

## Reporting summary

Further information on research design is available in the Nature Portfolio Reporting Summary linked to this article.

## Data availability

The fossil material of *Lijiangosaurus yongshengensis* can be checked on reasonable request under the permission from the Yunnan Biantun Cultural Museum of Yongsheng. The datasets for the phylogenetic analyses and ancestral state reconstruction are available as Supplementary Data and can be downloaded at Figshare[90].

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

## Acknowledgements
Specimen was prepared and protected by Jinzhao Ding, Wei Zhang, and Wei Gao; the artwork of the Middle Triassic nothosaurs from southwestern China was reconstructed by Kelai Li. We thank Xijun Ni, Tao Deng, Xing Xu, and Pingbo Cai for their guidance in accessing the specimen and the outcrop in the field. W.W. thanks Erin E. Maxwell, Rainer R. Schoch, Dayong Jiang, Lijun Zhao, and Torsten M. Scheyer for their hospitality during his visit to the collections; Zhonghe Zhou for the discussion to conceive the study. The Willi Hennig Society is acknowledged for making TNT publicly available. This research was supported by the Second Comprehensive Scientific Expedition on the Tibetan Plateau (2019QZKK0705), National Natural Science Foundation of China (42472023, 42002019), Youth Innovation Promotion Association of the Chinese Academy of Sciences, and Strategic Priority Research Program (B) of the Chinese Academy of Sciences (XDB26000000).

## Author contributions
W.W., Q.S., and C.L. conceived the research. Q.S., J.W., H.Z., and W.W. conducted the field work. Q.S. investigated the geological background. W.W. and Q.S. interpreted and described the anatomical morphology. W.W. performed the phylogenetic analyses and the ancestral state reconstruction. W.W. and Q.S. wrote the manuscript and prepared the figures. All the authors reviewed and discussed the manuscript and supplementary material.

## Competing interests
The authors declare no competing interests.
