## [Transparent Peer Review file · Communications Biology]

Earliest Long-necked Sauropterygian *Lijiangosaurus yongshengensis* and Plasticity of Vertebral Evolution in Sauropterygian Marine Reptiles

Corresponding Author: Dr Wei Wang

Version 0:

Reviewer comments:

Reviewer #1

(Remarks to the Author)
Review of COMMSBIO-24-2682-T

Dear editor, dear authors

I have revised the paper with my PhD student working on Triassic marine reptiles. While we are both convinced that this specimen represents a novel taxon with an unusual morphology, we feel that the paper is not ready for publication. There are a series of issues, most notably with the diagnosis, description, phylogenetic analysis; text is often imprecise and the fact that other sauropterygians with a long-necked existed in the Triassic is often toned down. Here are the main issues we would like to bring in the review:

- the skeleton is described in two places with lots of similarities: in the main text and in the supplementary information. Some features discussed in the main text as important synapomorphies are only found in the supplementary description and vice-versa.

- the text is often imprecise and partially incorrect. There are mentions of “stem” groups in fully extinct clades, many superlatives, and bones described as “invisible”. There is, at times, a confusion between synapomorphies and convergences and between clades and grades. In multiple places, the text states that an elongated neck was the reason for the success of plesiosaurians. There is no research backing this.

- the diagnosis is not useful: indeed, you classify this taxon as a nothosauria (L72) therefore, the diagnosis should not focus on distinguishing it from non-nothosaurians but should rather distinguish it from other nothosaurians. Please also clearly indicate which features are autapomorphies and which are shared with other nothosaurians.

-the phylogenetic dataset used is not ideal to test the precise position of the new taxon within eosauroptrygians: indeed, the dataset used was designed to analyse the phylogenetic relationships of Sauropterygiformes, containing only five nothosauroids for example. On the contrary, the dataset of Hu et al. (2024) contains much more (13). We also suggest to alter the analytical protocol: compute branch support metrics and employ (perhaps complementarily) implied weighting instead of just equal weighted maximum parsimony.

-Furthermore, the new taxon is recovered as closely related to Wangosaurus. Wangosaurus is a taxon with unstable phylogenetic placement; some recent papers regard it as a pistosauroid rather than a nothosaurian. Even the paper from which the dataset has been gathered (Wang et al 2022) regard Wangosaurus as having “unsolved” phylogenetic position. You yourself classify Wangosaurus as a “questionable pistosaur” in L289. This is crucial because a large paper of the paper focusses on the fact that Lijiangosaurus has a body shape unusual for nothosaurians. Its placement within this clade must

therefore be thoroughly tested because making these inferences. As mentioned above, testing the position of both Wangosaurus and Lijiangosaurus in a dataset thoroughly sampling nothosaurians and pistosaurians would be sounder.

- taxa are said to be convergent in multiple places of the paper, but without any statistical test. Convergence is usually assessed using the Stayton metrics, but other methods exist (e.g. Castiglione et al 2019 PLoS One).

-The fossil is highly incomplete posterior to the coracoid; yet you infer the length of the torso, the length and width of the tail. This is impossible to know and should be scored as "NA". Moreover, you mention that PC1 (86% of the total variance!) is "more about body sizes" and is ignored in the main text but you did not attempt any regression or allometry analysis to test this. Moreover, when using absolute sizes, you cannot rule out the fact that of the important morphological signal will be confounded in PC1 and that size signal can be found in other PCs as well. This is why it is much better to employ dimensionless ratios to conduct a PCA/PCoA/NMDS.

We also add a series of comments in a .pdf copy of the paper.

All the very best,

Valentin Fischer
Antoine Laboury

Reviewer #2

(Remarks to the Author)

The manuscript titled "Discovery of the Earliest Long-necked Sauropterygian and the Plasticity of Vertebral Evolution in Sauropterygian Marine Reptiles" includes an exhaustive description of an important fossil specimen. The new taxon is extremely valuable in improving our understanding of the early evolution of sauropterygians – a highly successful group of marine reptiles. It provides us with valuable data concerning the diversity and vertebral plasticity of these animals, in the crucial time of their evolutionary history – the early Middle Triassic. Thus, the manuscript constitutes an important contribution to palaeontology and will influence thinking in the field

In my opinion, the authors have described and illustrated the osteology of the specimen well, and their comparisons with other taxa and anatomical interpretations are also on point. Their conclusions are original and well defined, their claims are convincing and generally well supported. The figures include all the necessary data. The rules of describing new taxa are fulfilled, including the ZooBank reference. The phylogenetic analysis is inclusive and detailed. Thus, I assess the presented scientific value of the manuscript as high and promising, but several issues still have to be resolved to reach its full potential:

1. The whole manuscript and the supplementary information should be revised linguistically. While the general language of the contribution is fully understandable, a plethora of grammatical, logical and semantic errors can be found throughout the text, starting with the first sentence of the abstract (sic!). Due to the number of needed corrections, I refrain myself (as a non-native speaker myself) from pointing them all out. The authors should be more mindful of the meaning of some words (e.g., use "basal" rather than "primitive");
2. The authors use sentences like "The cervical count of 18 in Corosaurus, the earliest known and basal-most pistosaur, implies a status inherited from its eosauroptrygian ancestor." or "Compared to the ancestral condition of diapsids developing six cervical vertebrae and the basal sauropterygiforms possessing about 15 cervical vertebrae, the cervical vertebral counts and the neck lengths have more or less increased in all eosauroptrygians". While these statements are not necessarily wrong, they are hard to prove and not very insightful. In contrary, using parsimony and phylogenetic bracketing, it can be assumed that the last common ancestor of all eusauroptrygians had more cervicals than Corosaurus (see Fig. 4). Thus, I suggest the authors to support their claims by employing the ancestral state reconstruction. The authors are already well informed on the vertebral counts in sauropterygians, thus performing such analysis (e.g., using maximum parsimony approach in Mesquite) to recover the ancestral dorsal and cervical counts should not be time consuming. Moreover, it would not only make the conclusions of the manuscript less subjective, but also broaden its impact and importance in the field.
3. Please explicitly mention the contents of the Supplementary Information in the main text of the manuscript.
4. While the delineation of what is considered 'Nothosauria' in this contribution is clear, the authors should refine their usage of 'pistosauria' (for the members of Pistosauroidea, as I understand). Plesiosaurs are pistosauroids, thus mentioning "pistosauroids including plesiosaurs" is unnecessary and redundant. I understand that the authors use pistosauria as a paraphyletic group of basal non-plesiosaurian pistosauroids, but they should either clearly state that in the manuscript, or (preferably) address them as "non-plesiosaurian pistosauroids".
5. In the Systematic paleontology section the authors mention 'Nothosauria Baur, 1889', following Rieppel 2000, I assume. Baur (1889) included solely the family name 'Nothosauroidae', and the authorship of 'Nothosauria' is dubious (referred to von Zittel or Broili by various works). I advise the authors to properly reformulate the definition of the latter taxon, having in mind their new results. The authors already mention the key synapomorphies of the group, but a proper, well outlined, taxonomic emendation would be useful for future studies.
6. The "branch fracturing" (e.g., Fig. 3 between Sauropterygia and Eosauroptrygia) in the time-calibrated trees is highly unadvised. I understand its role in the data presentation, yet it is not feasible if we want to showcase not only the phylogenetic relationships of the taxa, but (more importantly) their evolutionary history as well.
7. The PCA methodology used by the authors is, in my opinion, not very insightful, for several reasons. Firstly, the measurement data used for the new taxon are largely estimated (for the trunk and tail), secondly, in my opinion the authors should use correlation-based PCA, instead of variance-covariance (as the data was gathered in uniform units), thirdly the

principal components 2 and 3 used by the authors cover only marginal variance loads (11% in sum), and, finally, I was not able to fully replicate the results provided by the authors, with the PC values being very slightly different (PC1=85vs86% and PC2=8v9%). To improve the scientific quality of the results obtained by the authors, I would advise either excluding PCA from the manuscript (it is barely commented on anyway) or use another kind of data standardization (e.g., Z-score Normalization) to reduce the impact of specimen size on PCs variance loads.

8. Personally, I find the potential ecological implications of study underexplored. We now know that the extreme neck elongation evolved convergently at least four times in the Anisian reptiles (trachelosaurids, tanystropheids, nothosaurs and pistosaurs). What was the cause of this never unparalleled similarity of forms across different contemporaneous lineages? Adding *Dinocephalosaurus* and *Tanystropheus* to the PCA matrix could also provide some interesting insights, as some other non-sauroptrygian aquatic reptiles are already included anyway.

Taking into consideration the mentioned advantages and disadvantages of the manuscript, as well as the content of the more specific comments included within the attached file, I think that the presented work can potentially be a valuable contribution to *Communications Biology*, but only after a major revision.

Reviewer #3

(Remarks to the Author)

The manuscript describes an interesting, because unexpectedly long-necked new specimen of a new nothosaur taxon. In this regard the special articulation of the cervical column is highlighted and compared to other accessory intervertebral articulations (AIA).

However, the authors missed an important paper which must be included into the comparison of the new specimen and other nothosaurs and into the discussion of AIAs, because it also shows a very special/unique AIA.

Klein et al. 2022

The redescription of the holotype of *Nothosaurus mirabilis* (Diapsida, Eosauroptrygia)—a historical skeleton from the Muschelkalk (Middle Triassic, Anisian) near Bayreuth (southern Germany) freely available under this link

<https://peerj.com/articles/13818/>

The manuscript needs a thorough correction of language/engl. There are several grammatical mistakes as well as sentences that are not or only hard to understand.

The description of teeth/dentition, the lower forelimb elements, and of the entire hindlimb is missing.

The humerus looks, contrary to the authors statement, not very nothosaur-like. In addition to the Bickelmann and Sander study, there are newer and more important references which can also be found in the above cited paper.

In the description, the total length of the new specimen is given with 2.5 m, in the discussion with 3 m, which both are not representing a very large individual, given a total length of *N. giganteus* and others of 6 m or even more.

Line 389: which species of *Neusticosaurus* ? or all 3 spec. ? what is with *Serpianosaurus* and *Prosantosaurus*

Nothosaurs are not in the "ancestral line" (line 255-256) of plesiosaurs ! please correct and rephrase.

The manuscript needs a thorough review of language, content and discussion, so I can only recommend MAJOR Revision.

Version 1:

Reviewer comments:

Reviewer #1

(Remarks to the Author)

See attached review

Reviewer #2

(Remarks to the Author)

The main points of my review have been taken into account by the Authors. My minor comments included in the submitted manuscript have also been addressed. Overall, I am satisfied with the Authors' approach to my (hopefully constructive) criticism. I think that the manuscript has benefited from the reviews, and I appreciate that the Authors took their time to perform the ancestral state reconstruction and another phylogenetic analysis. Some minor issues persist:

- Please read through the Supplementary Manuscript again and correct all the minor mistakes that are still contained within it (e.g. third sentence "Although the posterior dorsal, the sacral, and the proximal and distal [CAUDAL, I assume] vertebrae are missing possibly due to weathering [...]"; the explanation of 'cqp' marked on the Figure S3 is missing etc.);
- Lines 42-43 in the main text: "[...] the only exceptions convergently resembling plesiosaurs but with completely different

cervical shapes are a few bizarre early-diverging archosauromorphs [...]." Once more, this is not true. Archosauromorphs are not the ONLY exceptions, due to Hyphalosaurus not being an archosauromorph. Please reformulate (e.g. "Some reptile lineages (e.g. tanysaurians) have achieved a general bauplan similar to that of plesiosaurs, but with completely different vertebral morphology (citation).");

- Line 75 Keichousaurus instead of "Kechousaurus";

- Lastly but importantly, I was still not able to reproduce the results of the PCA. After log-transforming the supplied data, and performing both correlation or variance-covariance based PCA in Past 4.15, the PC variance yields differ, in both cases, from those provided in the manuscript. I assume that some miniscule changes have been made to the supplementary table after the PCA had been carried out. These are very minor differences (1% point), but nevertheless this issue should be addressed by the Authors to improve the quality of their work, especially if they insist on keeping the PCA within the main text, despite the concerns shared by me and at least one of the other Reviewers.

To reiterate, I think that the manuscript's quality has significantly improved and the remaining issues can be easily solved. I congratulate the Authors on this important publication and hope to see it published soon!

Best regards,
Adam Rytel

Reviewer #3

(Remarks to the Author)

Version 2:

Reviewer comments:

Reviewer #3

(Remarks to the Author)

Reviewer #4

(Remarks to the Author)

Lines 243-246. "30 cervical vertebrae, contrary to what was previously suggested, is no longer a synapomorphy for pistosaurs or plesiosaurs after our discovery of Lijiangosaurus". This statement is untenable based on your tree topology and ancestral states analysis, both of which imply an independent homoplastic acquisition of increased cervical number in Lijiangosaurus, but retain pistosaurs+plesiosaurs as a discrete monophyletic clade that presumably still shares increased cervical number as a synapomorphy. You must include the apomorphy lists for at least this key node in the tree to demonstrate whether there has actually been a character suite change involving loss of this shared feature.

Lines 256-258: This entire paragraph is awkwardly written and repeats information from earlier in the text. I suggest deleting it and beginning the section with "in the ventral view of the skull" from line 269.

Lines 314-315: You might also wish to note that other recent Bayesian phylogenies have resolved Wangosaurus as a nothosaurid (see Kear et al. 2024 Curr Biol 34, R553–R563).

Lines 372-374: This sentence integrates conflicting statements. Lijiangosaurus does not record "the first independent origin of the remarkably long neck" "in the whole sauropterygian evolutionary history" since increasing cervical numbers demonstrably occur in different plesiosauroid clades (e.g., cryproclidids, microcleidids, and elasmosaurids). This seems to be acknowledged in the second part of the sentence with "being earlier than the arising of pistosaurs and descendant plesiosaurs". What are the authors trying to say? Please clarify and emend accordingly.

Lines 374-377: "Since the function of the long neck in plesiosaurs and their pistosaur ancestors is undetermined, the long neck reported here provides evidence indirectly supporting that the neck elongation in nothosaurs and pistosaurs seems beneficial for foraging behavior". How? This statement is ambiguous and needs to be explained in more detail.

Reviewer #5

(Remarks to the Author)

The authors have dealt with the various minor and major criticisms raised by the Reviewers, especially Reviewer 1, to make sure statements in the text are accurate and clear. I have no further comments.

Version 3:

Reviewer comments:

Reviewer #5

(Remarks to the Author)

Reviewers' comments (Q) and Authors' response (A)

Dear editor and reviewers,

Thank you very much for considering our manuscript reporting a new nothosaur taxon and its implication on the early evolution of the dominant marine reptilian group, sauropterygians. We found the comments of reviewers are professional and generally positive on our discovery, but there are many constructive suggestions and criticisms on our morphologic comparisons, phylogeny, and other analyses, which are appreciated and accordingly modified in the revision by all of us. The specific comments on the PDF are respectively replied there and revised in the main manuscript, figures, and supplementary material. Here is our response to the main concerns of the three reviewers.

Review of COMMSBIO-24-2682-T

Reviewer #1 (Remarks to the Author):

Dear editor, dear authors

I have revised the paper with my PhD student working on Triassic marine reptiles. While we are both convinced that this specimen represents a novel taxon with an unusual morphology, we feel that the paper is not ready for publication. There are a series of issues, most notably with the diagnosis, description, phylogenetic analysis; text is often imprecise and the fact that other sauropterygians with a long-necked existed in the Triassic is often toned down. Here are the main issues we would like to bring in the review:

A: Thank you and your student for the professional and detailed comments, and your agreement on this novel sauropterygian taxon developing unexpected morphology, which we hope to have implications on our understanding of the early evolution of Triassic eosauroptrygians, as well as their descendant plesiosaurs.

Q- the skeleton is described in two places with lots of similarities: in the main text and in the supplementary information. Some features discussed in the main text as important synapomorphies are only found in the supplementary description and vice-versa.

A: We have re-organized some of the description in both main text supplementary information (SI). The crucial features such as the morphology of the palate, the dentition, the humerus, and the pes are removed and summarized from the SI to the main text. Some specific modifications are replied to in the PDF.

Q- the text is often imprecise and partially incorrect. There are mentions of "stem" groups in fully extinct clades, many superlatives, and bones described as "invisible". There is, at times, a confusion between synapomorphies and convergences and between clades and grades. In multiple places, the text states that an elongated neck was the reason for the success of plesiosaurians. There is no research backing this.

A: The definition of the "stem" group of Sauropterygia follows that in Rieppel, 2000, which is not identified according to the extinct and extant lineages as commonly used but accepted and used by many researchers working on Triassic sauropterygians. The improper phrases and expressions like some superlatives and "invisible", as well as the confusion of clades and grades are one-by-one modified according to the reviewers' comments, for which please find

our reply in the PDF. As for “the long neck beneficial to the success of plesiosaurs”, we agree that there is no clear evidence for that, and have omitted it in the abstract, and weakened the related sentence as a possible case in the discussion.

Q- the diagnosis is not useful: indeed, you classify this taxon as a nothosauria (L72) therefore, the diagnosis should not focus on distinguishing it from non-nothosaurians but should rather distinguish it from other nothosaurians. Please also clearly indicate which features are autapomorphies and which are shared with other nothosaurians.

A: Thanks for this suggestion. Our previous version of diagnosis wrote the characters different from non-nothosaurian first before the autapomorphies compared with other nothosaurian. In the revision, we have revised the diagnosis by first putting the autapomorphies of *Lijiangosaurus yongshengensis* distinguishing from other nothosaurians, added more detailed comparisons, and kept the differences from non-nothosaurians too but at the end of the diagnosis, which we think is still useful.

Q-the phylogenetic dataset used is not ideal to test the precise position of the new taxon within eosauropterygians: indeed, the dataset used was designed to analyse the phylogenetic relationships of Sauropterygiformes, containing only five nothosauroids for example. On the contrary, the dataset of Hu et al. (2024) contains much more (13). We also suggest to alter the analytical protocol: compute branch support metrics and employ (perhaps complementarily) implied weighting instead of just equal weighted maximum parsimony.

A: This is a good criticism, which we have paid much attention to and spent some time on. In fact, on the genus level, the dataset of Hu et al. (2024) only contains four possible nothosauroid genera (*Simosaurus*, *Germanosaurus*, *Nothosaurus*, and *Lariosaurus*), although there are many species of *Nothosaurus* and *Lariosaurus*. Notably, *Ceresiosaurus* has already been considered as the junior synonym of *Lariosaurus* (Rieppel, 1998), and *Paranothosaurus* as *Nothosaurus giganteus* (Rieppel, 2000), which are still used with the outdated taxon names without any explanation from Hu et al. On the contrary, the dataset from Wang et al. (2022) employed in this study contains eight possible nothosauroid genera (*Simosaurus*, *Paludidraco*, *Brevicaudosaurus*, *Germanosaurus*, *Lijiangosaurus*, *Wangosaurus*, *Lijiangosaurus*, *Nothosaurus*, *Lariosaurus*).

We carefully compare the different recent matrices focusing on sauropterygians from different authors, and briefly summarize here:

(1) Hu et al. 2024 (203 characters, 43 taxa with 13 *N.* and *L.* spp.) from Li et Liu 2020 (181 characters, 36 taxa with 14 *N.* and *L.* spp. including *L. sanxiaensis* as a possible synonym of *Hanosaurus*) combined from Liu et al. 2011 (137 characters, 29 taxa) and Liu et al. 2014 (74 characters, 22 taxa with 19 *N.* and *L.* spp.) originally from Rieppel 2002;

(2) Wang et al. 2022 (181 characters, 62 taxa with 42 sauropterygian genera) mainly from Neenan et al. 2013 (140 characters, 43 taxa with 21 sauropterygian taxa) largely modified from Liu et al. 2011 and originally from Rieppel 2002;

(3) Xu et al. 2023 (183 characters, 63 taxa with 10 *N.* and *L.* spp.) similar to Xu et al. 2022 from Lin et al. 2021 (148 characters, 49 taxa with 21 *N.* and *L.* spp.) from Lin's PhD thesis 2019 (only published in Chinese) probably from Liu et al. 2014 and originally from Rieppel 2002.

All these current matrices share the same root from Rieppel 2002 but show many differences

in character identification and taxon sampling. The dataset from Hu et al. is the latest, and that from Xu et al. is later than Wang et al. 2022. However, we do not consider the former two as the most adequate ones for resolving a novel sauropterygian, even an eosauropterygian, because they are designed more to test the interrelationships of multiple species within *Nothosaurus* and *Lariosaurus*, as discussed in Li and Liu 2020 and Liu et al. 2014, by involving many species of these two genera. By comparing the character lists between Hu et al. 2024 and Wang et al. 2022, we notice that many characters in Hu et al. 2024 are originally used to distinguish different species within *Nothosaurus*, for example, ratios on cranial structures, which are hard to be homologous within larger sampling involving more and more eosauropterygians and other sauropterygians. Even though the matrix in Wang et al. 2022 seems designed for a larger group, Sauropterygiformes, we find the dataset contains a broader sampling of sauropterygians including eosauropterygians with relatively more reasonable characters/states and out groups as the authors discussed in Wang et al. 2022, and therefore, it is suitable for testing the position of *Lijiangosaurus* too. By adding more very recently discovered taxa, we constructed the matrix in this study of *Lijiangosaurus*, which includes 181 characters and 68 taxa with 48 sauropterygian taxa (almost all hitherto known sauropterygian genera).

For further feedback to the reviewers, we also conducted analysis by scoring *Wangosaurus*, *Lijiangosaurus*, *Brevicaudosaurus* based on our personal observation in the matrix of Hu et al. 2024 to test the topology, and the result (Bootstrap values >5 above and Bremer supports below each node) is attached here and provided in our revised SI with the updated matrix and scores of *Wangosaurus*, *Lijiangosaurus*, and *Brevicaudosaurus*.

Reference of the different matrices:

- Rieppel, O. The status of the sauropterygian reptile genera *Ceresiosaurus*, *Lariosaurus*, and *Silvestrosaurus* from the Middle Triassic of Europe. *Fieldiana, Geology, new ser.*, no. 38 (1998).
- Rieppel, O. *Handbook of Paleoherpétology*, 12A (Verlag Dr Friedrich Pfeil), pp. 1–134 (2000).
- Hinz, J. K., Matzke, A. T. & Pfretzschner, H.-U. A new nothosaur (Sauropterygia) from the Ladinian of Vellberg–Eschenau, southern Germany. *Journal of Vertebrate Paleontology* 39, e1585364 (2019).
- Hu, Y. W., Li, Q. & Liu, J. A new pachypleurosaur (Reptilia: Sauropterygia) from the Middle Triassic of southwestern China and its phylogenetic and biogeographic implications. *Swiss Journal of Palaeontology* 143(1), 1 (2024).
- Li, Q., & Liu, J. (2020). An Early Triassic sauropterygian and associated fauna from South China provide insights into Triassic ecosystem health. *Communications Biology*, 3(1), 63 (2020).
- Liu, J. et al. A gigantic nothosaur (Reptilia: Sauropterygia) from the Middle Triassic of SW China and its implication for the Triassic biotic recovery. *Sci. Rep.* 4, 7142 (2014).
- Liu, J., Rieppel, O., Jiang, D. Y., Aitchison, J. C., Motani, R., Zhang, Q. Y., Zhou, C. Y., & Sun, Y. Y. A new pachypleurosaur (Reptilia, Sauropterygia) from the lower Middle Triassic of SW China and the phylogenetic relationships of Chinese pachypleurosaur. *Journal of Vertebrate Paleontology*, 31(2), 292–302 (2011).
- Rieppel, O., P. M. Sander, and G. W. Storrs. 2002. The skull of the pistosaur *Augustasaurus* from the Middle Triassic of northwestern Nevada. *Journal of Vertebrate Paleontology* 22:577–592.
- Wang, W., Shang, Q., Cheng, L., Wu, X. C. & Li, C. Ancestral body plan and adaptive radiation of sauropterygian marine reptiles. *iScience* 25, 105635 (2022).
- Neenan, J., Klein, N. & Scheyer, T. European origin of placodont marine reptiles and the evolution of crushing dentition in Placodontia. *Nat. Commun.* 4, 1621 (2013).
- Xu, G. H. et al. A new long-snouted marine reptile from the Middle Triassic of China illuminates pachypleurosauroid evolution. *Sci Rep* 13, 16 (2023).
- Xu, G.-H., Ren, Y., Zhao, L.-J., Liao, J.-L. & Feng, D.-H. A long-tailed marine reptile from China provides new insights into the Middle Triassic pachypleurosaur radiation. *Sci. Rep.* 12, 7396 (2022).
- Lin, W. B. et al. *Panzhousaurus rotundirostris* Jiang et al., 2019 (Diapsida: Sauropterygia) and the recovery of the monophyly of Pachypleurosauridae. *J. Vertebr. Paleontol.* 41, e1901730 (2021).
- Lin, W. B., M. Zhou, and D.-Y. Jiang. Systematic study of the eosauroptrygians from the Triassic of South China. Science Press, Beijing, China, 154pp (2019) (in Chinese)

Q -Furthermore, the new taxon is recovered as closely related to *Wangosaurus*. *Wangosaurus* is a taxon with unstable phylogenetic placement; some recent papers regard it as a pistosauroid rather than a nothosaurian. Even the paper from which the dataset has been gathered (Wang et al 2022) regard *Wangosaurus* as having “unsolved” phylogenetic position. You yourself classify *Wangosaurus* as a “questionable pistosaur” in L289. This is crucial because a large paper of the paper focusses on the fact that *Lijiangosaurus* has a body shape unusual for nothosaurians. Its placement within this clade must therefore be thoroughly

tested because making these inferences. As mentioned above, testing the position of both *Wangosaurus* and *Lijiangosaurus* in a dataset thoroughly sampling nothosaurians and pistosaurians would be sounder.

A: Thank you for this constructive suggestion about *Wangosaurus*, which we largely agree with. In the phylogenetic result, we consider *Wangosaurus* as the sister group of *Lijiangosaurus*, because it is what our result indicates and we would like to objectively describe the result. In the discussion part, we are always cautious about the position of *Wangosaurus* without intensive study specifically on this taxon. It was written as “questionable pistosaur” because of its unsolved position as you said and we wrote in our manuscript. However, *Wangosaurus* being either a nothosaur or a pistosaur does not matter to the nothosaurian affiliation of *Lijiangosaurus*, since the morphology and phylogeny support *Lijiangosaurus* is a nothosaur. Please check our additional result using the matrix in Hu et al. (2024), where *Lijiangosaurus* and *Wangosaurus* are resolved as sister groups too within nothosaurs.

Here we attach our unpublished photo of the cranial ventral side of *Wangosaurus brevirostris* (holotype, GMPKU-P-1529) for your reference, and the scores of this taxon in the dataset are based on our personal observation.

[image redacted]

Q- taxa are said to be convergent in multiple places of the paper, but without any statistical test. Convergence is usually assessed used the Stayton metrics, but other methods exist (e.g. Castiglione et al 2019 PLoS One).

A. Thank you for this suggestion to more quantitatively demonstrate the convergence. Stayton metrics and the method introduced in Castiglione et al. (2019 PLoS One) are efficient and powerful. We have read some of the publications of Stayton and that of Castiglione et al., and tried to understand and practice, but due to our limited knowledge and experience with such statistical techniques, we feel sorry that we cannot employ these methods in this current study and hope to try in our future project cooperating with experts on these methods like you. As we read, there are also many other publications on newly found fossils that could mention convergence only based on the phylogeny without such a comprehensive statistical method when the authors discuss morphological comparisons. Considering the convergence discussed in this manuscript is mainly about one character, the long neck reflected by the number of cervical vertebrae, alternatively, as another reviewer suggested, we employed the ancestral state reconstruction about the cervical count. In our result added to this revision

(revised Fig. 4), the common ancestor of *Lijiangosaurus*, *Wangosaurus*, and pistosaurs developed a neck with no more than 25 cervical vertebrae, which indicates that the clade including *Lijiangosaurus* and *Wangosaurus* evolved elongate necks (more than 30 cervical vertebrae) independently compared to those in pistosaurs. This ancestral state reconstruction shows more phenotypic resemblance on cervical count in descendants than their ancestors, which is convergent. We appreciate your understanding and hope for our potential collaboration on such macroevolution on marine reptiles.

Q-The fossil is highly incomplete posterior to the coracoid; yet you infer the length of the torso, the length and width of the tail. This is impossible to know and should be scored as "NA". Moreover, you mention that PC1 (86% of the total variance!) is "more about body sizes" and is ignored in the main text but you did not attempt any regression or allometry analysis to test this. Moreover, when using absolute sizes, you cannot rule out the fact that of the important morphological signal will be confounded in PC1 and that size signal can be found in other PCs as well. This is why it is much better to employ dimensionless ratios to conduct a PCA/PCoA/NMDS.

A: Yes, the fossil is incomplete posterior to the anterior dorsal vertebrae. However, the incomplete skeleton is preserved *in situ* on the same layer on a large block of limestone, and the length of the torso could be directly measured from the first dorsal vertebra to the sacrum that is marked by the sacral rib and the pubis (blue line in the image). We estimated the tail length from the sacrum to the last articulated caudal vertebra adding the length of all preserved isolated possible caudal vertebrae (purple line in the image), and estimated the maximum trunk width by doubling the preserved longest dorsal rib, and the maximum tail width by doubling the length of the sacral rib. These estimations are rough, but they are not entirely unreliable and could not be more exact considering such a large fossil individual. Hence, we attempt to use the data of this new specimen in the PCA, and hope for your understanding of the "poor quality" of some estimated measurements in this incomplete specimen.

[image redacted]

As for the PC1 and the signal from size, we fully agree with you. We have thought about employing dimensionless ratios, but every ratio is influenced by two measurements, for

example, a large ratio between neck length and skull length could be resulted by either a long neck or a small skull. We would like to learn more methods from publications (like some of yours) to investigate the morphospace in Triassic sauropterygians in our future study, when we suggest keeping the PCA results as they are in this current study, which could show the overlap of *Lijiangosaurus*, *Wangosaurus*, and pistosaurs on body shapes. According to your comments, we have removed the PC1-PC2 figure to the main text as a part of the revised Fig. 5.

We also add a series of comments in a .pdf copy of the paper.

A: Thank you a lot for your thorough comments and suggestions, which are replied in the PDF.

All the very best,

Valentin Fischer
Antoine Laboury

Reviewer #2 (Remarks to the Author):

The manuscript titled “Discovery of the Earliest Long-necked Sauropterygian and the Plasticity of Vertebral Evolution in Sauropterygian Marine Reptiles” includes an exhaustive description of an important fossil specimen. The new taxon is extremely valuable in improving our understanding of the early evolution of sauropterygians – a highly successful group of marine reptiles. It provides us with valuable data concerning the diversity and vertebral plasticity of these animals, in the crucial time of their evolutionary history – the early Middle Triassic. Thus, the manuscripts constitutes an important contribution to palaeontology and will influence thinking in the field

In my opinion, the authors have described and illustrated the osteology of the specimen well, and their comparisons with other taxa and anatomical interpretations are also on point. Their conclusions are original and well defined, their claims are convincing and generally well supported. The figures include all the necessary data. The rules of describing new taxa are fulfilled, including the ZooBank reference. The phylogenetic analysis is inclusive and detailed. Thus, I assess the presented scientific value of the manuscripts as high and promising, but several issues still have to be resolved to reach its full potential:

A: Thank you for these positive comments and your thorough review of our work.

Q1. The whole manuscript and the supplementary information should be revised linguistically. While the general language of the contribution is fully understandable, a plethora of grammatical, logical and semantic errors can be found throughout the text, starting with the first sentence of the abstract (sic!). Due to the number of needed corrections, I refrain myself (as a non-native speaker myself) from pointing them all out. The authors should be more mindful of the meaning of some words (e.g., use “basal” rather than “primitive”);

A: Thank you for pointing out these lingual deficiencies as other reviewers mention too. We have corrected most of the errors since other reviewers added comments on the PDF, and tried to rephrase some sentences with the help of a native English speaker. Many words are changed to be more precise, including replacing “primitive” with “basal”.

Q2. The authors use sentences like “The cervical count of 18 in *Corosaurus*, the earliest known and basal-most pistosaur, implies a status inherited from its eosauropterygian ancestor.” or “Compared to the ancestral condition of diapsids developing six cervical vertebrae and the basal sauropterygiforms possessing about 15 cervical vertebrae, the cervical vertebral counts and the neck lengths have more or less increased in all eosauropterygians”. While these statements are not necessarily wrong, they are hard to prove and not very insightful. In contrary, using parsimony and phylogenetic bracketing, it can be assumed that the last common ancestor of all eusauropterygians had more cervicals than *Corosaurus* (see Fig. 4). Thus, I suggest the authors to support their claims by employing the ancestral state reconstruction. The authors are already well informed on the vertebral counts in sauropterygians, thus performing such analysis (e.g., using maximum parsimony approach in Mesquite) to recover the ancestral dorsal and cervical counts should not be time consuming. Moreover, it would not only make the conclusions of the manuscript less subjective, but also broaden its impact and importance in the field.

A: This is a constructive suggestion that we appreciate a lot and fully agree. As one focus of this study is about the evolution of the cervical counts (not dorsal counts since it is unknown in this new taxon) in Triassic eosauropterygians with the cervical vertebra number in most of the Triassic eosauropterygian taxa available, we conducted the ancestral state reconstruction by maximum parsimony in Mesquite v. 3.81, and modified Fig. 4 with this result and added this employment in the Method. This ancestral state reconstruction, as assumed by the reviewer, not only explains the variations of cervical counts more quantitatively and clearly than our previous description, but also provides the evaluations of cervical states of the common ancestors on several nodes. Moreover, we rewrote the sentence about *Corosaurus* to be “the cervical count of 18 in *Corosaurus*, the earliest known and basal-most pistosaur, implies a status inherited and even reduced from its eosauropterygian ancestor.”

Q3. Please explicitly mention the contents of the Supplementary Information in the main text of the manuscript.

A: On the one hand, we rearranged the osteological description of both the main text and the Supplemental Information to ensure any crucial diagnosis and description present in the main text, as other reviewers also suggested. On the other hand, we went through the main text to check that all the contents, especially the supplemental figures, have been properly mentioned and referred to.

Q4. While the delineation of what is considered 'Nothosauria' in this contribution is clear, the authors should refine their usage of 'pistosauria' (for the members of Pistosauroida, as I understand). Plesiosaurs are pistosauroids, thus mentioning "pistosauroids including plesiosaurs" is unnecessary and redundant. I understand that the authors use pistosauria as a paraphyletic group of basal non-plesiosaurian pistosauroids, but they should either clearly state that in the manuscript, or (preferably) address them as "non-plesiosaurian pistosauroids".

A. To avoid confusing terminology, we previously tried to use as few but clear taxonomical terms as possible, but unfortunately, those appeared not precise enough to you and other reviewers. The clade, Pistosauroida, is monophyletic, and equals "pistosauroids" in this manuscript. We modified some of the text mentioning these names of groups, and consistently use "basal pistosauroids" referring to "non-plesiosaurian pistosauroids" in most cases, and use "pistosauroids" and "Pistosauroida" a few times when the taxonomy is emphasized. The identification of Pistosauroida is also shown in Fig 3. Additionally, we would like to remain the words "pistosauroids including plesiosaurs" once in the introduction for explanation in the earlier part of the text. We hope our revision can reduce readers' confusion.

Q5. In the Systematic paleontology section the authors mention ‘Nothosauria Baur, 1889’, following Rieppel 2000, I assume. Baur (1889) included solely the family name ‘Nothosauridae’, and the authorship of ‘Nothosauria’ is dubious (referred to von Zittel or Broili by various works). I advise the authors to properly reformulate the definition of the latter taxon, having in mind their new results. The authors already mention the key synapomorphies of the group, but a proper, well outlined, taxonomic emendation would be useful for future studies.

A. Thank you for this careful investigation of the authorships. We double-checked that “Nothosauria Baur 1889” was listed by Rieppel (2000, fig. 1 and p. 64). Your suggestion of reformulating the definition of Nothosauria is great, and we have added this part in our revised Systematic paleontology, including the Emended definition, Composition, and Diagnosis of Nothosauria. The same content of synapomorphies are deleted from the phylogenetic result.

Q6. The “branch fracturing” (e.g., Fig. 3 between Sauropterygia and Eosauropterygia) in the time-calibrated trees is highly unadvised. I understand its role in the data presentation, yet it is not feasible if we want to showcase not only the phylogenetic relationships of the taxa, but (more importantly) their evolutionary history as well.

A. Yes, it is better to avoid such branch fracturing. We did that because the nodes were not time-calibrated and we attempted to save space and not to make some lines too short to read. In our revision, Fig. 3 has been modified without “branch fracturing” and to be clear at the same time.

[figure redacted]

Q7. The PCA methodology used by the authors is, in my opinion, not very insightful, for several reasons. Firstly, the measurement data used for the new taxon are largely estimated (for the trunk and tail), secondly, in my opinion the authors should use correlation-based PCA, instead of variance-covariance (as the data was gathered in uniform units), thirdly the

principal components 2 and 3 used by the authors cover only marginal variance loads (11% in sum), and, finally, I was not able to fully replicate the results provided by the authors, with the PC values being very slightly different (PC1=85vs86% and PC2=8v9%). To improve the scientific quality of the results obtained by the authors, I would advise either excluding PCA from the manuscript (it is barely commented on anyway) or use another kind of data standarization (e.g., Z-score Normalization) to reduce the impact of specimen size on PCs variance loads.

A: Thank you for these reasonable comments. As for the estimated measurements of this new specimen, because the skeleton is preserved *in situ* though incompletely, the length of trunk could be roughly measured from the first dorsal to the sacrum, and the tail length from the sacrum to the last articulated caudal vertebrae adding the lengths of isolated possible caudal vertebrae, and the width of trunk and tail are estimated by doubling the length of dorsal rib and sacral rib respectively, which could be not that unreliable considering such a large animal. The size also has signal on body shape and convergence of long-necked nothosaurians and pistosaurs, as other reviewers mentioned, and therefore, we decided not to use Z-score normalization on the measurements in this study, but thank you for this suggestion on data standardization that we will consider to use in our other studies. Given the comments from you and other reviewers, we prefer remaining the PCA results in our manuscript with figures of PC1-PC2 and PC2-PC3 together in the revised Fig. 5, since it more or less reflects the convergence by showing the short distances between the plots of long-necked nothosaurs and pistosaurs. If you still feel this PCA is needless in this study, we would like to put the results in the Supplemental Information.

Q8. Personally, I find the potential ecological implications of study underexplored. We now know that the extreme neck elongation evolved convergently at least four times in the Anisian reptiles (trachelosaurids, tanystropheids, nothosaurs and pistosaurs). What was the cause of this never unparalleled similarity of forms across different contemporaneous lineages? Adding *Dinocephalosaurus* and *Tanystropeus* to the PCA matrix could also provide some interesting insights, as some other non-sauropterygian aquatic reptiles are already included anyway.

A: Yes, we fully agree with you about comparing these long-necked taxa together. The cause of long-necked convergence is not easy to address, but it is an interesting topic to study. In our another separate project with a manuscript under preparation on the evolution of body plan in tanysaurians, we will compare the body shapes among different secondarily marine reptiles including the long-necked sauropterygians and tanysaurians, and we hope that research could provide further insights on this convergence to you and other colleagues.

Taking into consideration the mentioned advantages and disadvantages of the manuscript, as well as the content of the more specific comments included within the attached file, I think that the presented work can potentially be a valuable contribution to Communications Biology, but only after a major revision.

Reviewer #3 (Remarks to the Author):

The manuscript describes an interesting, because unexpectedly long-necked new specimen

of a new nothosaur taxon.

Q1. In this regard the special articulation of the cervical column is highlighted and compared to other accessory intervertebral articulations (AIA).

However, the authors missed an important paper which must be included into the comparison of the new specimen and other nothosaurs and into the discussion of AIAs, because it also shows a very special/unique AIA.

Klein et al. 2022

The redescription of the holotype of *Nothosaurus mirabilis* (Diapsida, Eosauropterygia)—a historical skeleton from the Muschelkalk (Middle Triassic, Anisian) near Bayreuth (southern Germany)

freely available under this link

<https://peerj.com/articles/13818/>

A: Thank you for reminding us of this relevant publication that should definitely be considered and cited in our study here, since it may not have been published when we were preparing the comparative description of our study. We have added the citation of Klein et al. (2022) about the redescription of *Nothosaurus mirabilis*. The comparisons between *Lijiangosaurus* and *N. mirabilis* have been added, including the body size, the height of dorsal neural spines, and especially the accessory intervertebral articulations (AIA). The AIA in *Lijiangosaurus* is lower than the pre-postzygapophysis interpreted as infrapre-infrapostzygapophysis and present on dorsal and anterior caudal vertebrae, while the AIA in *N. mirabilis* is upper than the pre-postzygapophysis and supersized on dorsal vertebrae, which is identified as zygosphenes-zygantrum and different from those in *Lijiangosaurus*. We fully agree with the identification and description by Klein et al. (2022), and have added this AIA in *N. mirabilis* in our discussion in the main text and comparison in Fig. 4.

Q2. The manuscript needs a thorough correction of language/engl. There are several grammatical mistakes as well as sentences that are not or only hard to understand.

A: Please excuse us as non-English speakers. We have corrected grammatical mistakes and paraphrased some confusing sentences following the suggestions from you and other reviewers, as well as a native English speaker at our institute. We hope that the revision becomes more clear and precise in language.

Q3. The description of teeth/dentition, the lower forelimb elements, and of the entire hindlimb is missing.

A: Because of the limitation of space of the main text (<5000 words), we did describe the dentition, as well as all the preserved elements of forelimbs and hindlimbs but put in the Supplemental Information (SI). Sorry for too little mention in the main text for readers to find the information. We have added more references to SI about the above anatomical parts in this revised main text.

Q4. The humerus looks, contrary to the authors statement, not very nothosaur-like. In addition to the Bickelmann and Sander study, there are newer and more important references which can also be found in the above cited paper.

A: We double-checked the references describing the humeral morphology of *Nothosaurus*, including Bickelmann and Sander (2008), and Nickel et al. (2022), which are cited in this study. *Lijiangosaurus* is not *Nothosaurus*, and therefore its humeral morphology can be different from the species of *Nothosaurus*, and there is even morphologic variation of humerus among different species of *Nothosaurus*. By writing “this humerus is again similar to that in nothosaurs” (line 225-226), we meant that the humerus of *Lijiangosaurus* is more similar to nothosaurians than pistosaurs as the evidence of nothosaurian but not pistosauroid affinity of *Lijiangosaurus*. We have modified this statement to be “this humerus is again more similar to those in nothosaurs than in pistosaurs and more basal eosauropterygians” for better clarity. Thanks for your understanding of our purpose.

Q5. In the description, the total length of the new specimen is given with 2.5 m, in the discussion with 3 m, which both are not representing a very large individual, given a total length of *N. giganteus* and others of 6 m or even more.

A: The preserved part of this new specimen is about 2.5 meters, and so we wrote the total body length “over 2.5 meters” in the Diagnosis and “approximately 3 meters” in the Discussion, which are not contradictory considering the incompleteness of this skeleton. Yes, however, you are right that it does not represent a very large individual among nothosaurians, when some *Nothosaurus* species are much larger, though this new specimen is bigger than most of the nothosaurians, including *Nothosaurus* and *Lariosaurus* spp. from south China. For more preciseness, we have omitted the statement of “large size” in our description and discussion on this specimen, and modified the sentence in Diagnosis to be “a medium-to-large-sized nothosaurian” and added the comparisons with *Nothosaurus giganteus* and *Nothosaurus mirabilis*.

Q6. Line 389: which species of *Neusticosaurus* ? or all 3 spec. ? what is with *Serpianosaurus* and *Prosantosaurus*

A: According to Fig. 15 in Carroll & Gaskill (1985), an isolated dorsal vertebra of “*Pachypleurosaurus*” *edwardsi* shows AIA, and therefore, it should be *Neusticosaurus edwardsi*. We are not able to know the situation in the other two species of *Neusticosaurus*, and so, following your suggestion, we added the species name “edwardsi” in Line 389. As for the AIA in *Serpianosaurus* and *Prosantosaurus*, we have not found evidence of AIA from our pictures and the literature, though they possibly also developed AIAs. Additionally, we mistakenly identified the AIA in *Neusticosaurus edwardsi* as hyposphene-hypantrum, which should be zygosphene-zygantrum, and we have corrected it in our revision (Line 431 and Fig. 4).

Q7. Nothosaurs are not in the "ancestral line" (line 255-256) of plesiosaurs! please correct and rephrase.

A: Line 255-256 is changed to Line 285-286, in which we wrote "Eusauroptrygia comprises two major clades (Fig. 3), Nothosauria and Pistosaurioidea, when the former is represented by *Nothosaurus*, and the latter shares several features with its descendant lineage, Plesiosauria." Yes, *Nothosaurus*/nothosaurs is/are not in the "ancestral line" or "ancestral grade" of plesiosaurs, but is/are another clade of eosauroptrygians. We have thoroughly checked our similar expression of this "ancestral line" or "ancestral grade" in the manuscript, and have modified them.

The manuscript needs a thorough review of language, content and discussion, so I can only recommend MAJOR Revision.

A: Thank you again for all your professional concerns. More comments on the PDF are replied there and accordingly revised in the text and figures.

Dear editors and reviewers,

Many thanks to the second round of reviews. We noticed that the second and the third referees had little or only a few suggestions, but the first referees still emphasized some concerns. On behalf of all co-authors, I explain our further modification and make specific response here.

Reviewers #1

Dear editor, dear authors,

We have reviewed this new version of the paper. Sadly, we see in the revised version as well as in the rebuttal letter that the authors did not wish to substantially modify the paper despite important points and errors we discussed during the first round of reviews.

We will detail the main points below, but as the authors do not wish to modify the paper and maintain erroneous terminologies and erroneous analytical protocols, we will again recommend rejection. We do not want to review it again unless it is substantially modified.

A: In the first round of reviews, we have carefully checked and addressed each concern from all the reviewers point-to-point, including the main criticisms in the letter and about 150 comments on the PDF from Prof Valentin Fischer and Dr Antoine Laboury. Many of their suggestions were accepted, incorporated, and revised in our revision. As for the parts we didn't modify substantially, in the last revision and response, we attempted to explain more about what we have done and written. However, regrettably, it seemed a rebuttal to the Reviewers #1, which was not our intention. We now are clear about the reasonable insists of you, and further revised our manuscript, and answered your major queries below.

1. Implications for plesiosaurian evolution

In the main text and in the rebuttal letter, it is said "we hope to have implications on our understanding of the early evolution of Triassic eosauroptrygians, as well as their descendant plesiosaurs."

This is an enduring misconception present in the paper: how can a peculiar morphology in a single nothosauroid have implications for the evolution of plesiosaurians (which do not descent from nothosaurs)? We understand the implications for eosauroptrygia as a whole, that it is indeed very interesting to have an earlier example of neck elongation in eosauroptrygians. Perhaps that is what you mean but then we would suggest to clearly state it in the abstract, intro, discussion, and conclusion.

A: It is unquestionable that plesiosaurs are not descent from nothosaurs but pistosaurs, which has been clearly introduced in our manuscript, such as in the abstract, the introduction, and Figure 3. A long neck with more than 30 cervical vertebrae is suggested to be a major feature or even a synapomorphic character in pistosaurs including the early plesiosaurs (e.g., Wintrich et al., 2017, Sci. Adv.), and has never been reported from any other subclade in sauropterygians before our discovery of *Lijiangosaurus*. This discovery of a new nothosaur with an unexpected long neck with 42 cervical vertebrae not only updates our knowledge of

the neck variation in eosauroptrygians in general, but also suggests that the long neck is not a characteristic morphology in pistosaurs/plesiosaurs and not a key innovation for their “unique” lifestyle and survival from the end-Triassic extinction. For another example, *Anchiornis* is neither a bird nor the direct ancestor of birds, but it has the implication on the evolution of feather and flight in dinosaurs (Hu et al., 2009, Nature). Thanks for your comments again for this issue, we did need and have done some modification through our manuscript to better state this implication in our manuscript as following listed.

L5, L48: it is said in the abstract and the introduction that the origin of the long neck in plesiosaurians is controversial. It is not true, it arose thanks to a combination of somitogenesis and differential growth and was already present, to a lesser extent, in early pistosauroids.

A: “The origin” in the abstract and introduction includes the selective pressure and preference of such a long neck in pistosaurs, which related to the advantage of developing a long neck. It has been clear that somitogenesis and differential growth contributed to the long neck, while this was the mechanism, as a part of the origin, of the long neck. Yes, we admit that “the origin” here is misleading to some extent, and have omitted “the origin” in these sentences in L5 and L48 (L47 in Revision2). The discovery of a long-necked eosauroptrygian (*Lijiangosaurus yongshengensis*) out of the clade of stem-group pistosaurs and plesiosaurs provides circumstantial evidence hypothetically supporting the foraging adaptation of the long neck in sauropterygians.

L51: once again, the authors are seemingly suggesting that things that are actually known, aren't. If the “hydrofoil bodyplan” they mention it the bodyplan with four long flippers, then it evolved concomitantly than a fairly long neck in pistosauroids.

A: Yes, this sentence in our manuscript was misleading too. We have modified it to be “compared to other eosauroptrygian groups that became extinct during the Late Triassic or even at the end of the Middle Triassic, such as pachypleurosaurs and nothosaurs, it is unclear whether the remarkably long neck of pistosaurs was a key evolutionary innovation that enabled their survival through the end-Triassic extinction, even though such a long neck is suggested to be hydrodynamically disadvantageous for their locomotion”, and deleted the statement about its hydrofoil body plan to focus on the neck (see L45-L48 in Revision2).

Other modifications in Introduction and Discussion about the issue of the implication for neck elongation in plesiosaurs:

L59: We deleted the last sentence “emphasizes the importance of key fossil discoveries for evaluating the macroevolution in marine reptiles”, because compared with other studies focusing on macroevolution of sauropterygians, we did little analysis about their macroevolution.

L378: We deleted the statement about “hydrodynamic advantages as previously suggested by some researcher”, when the hydrodynamic disadvantage has been proved by some recently published paper, which is also cited in our manuscript.

L379: We added “it seems that rather than the long neck, the locomotion and pelagic life style are more crucial evolutionary novelties in plesiosaurs”, and cited the review paper of long necks in plesiosaurs here.

2. The use of incorrect phylogenetic terms.

In the rebuttal letter, the authors advocate for the use of “stem” in Sauropterygia and indicate that a crown group exist as well (L23-25). This is a blatant error. A crown group can only be defined by extant taxa, and the lineages basal to the crown is the paraphyletic stem group. We really cannot understand why the authors are willing to use an erroneous terminology because it was mentioned in a paper 24 years ago.

A: As we explained, the definition of “stem group” and “crown group” of sauropterygians follows Dr Olivier Rieppel, which can be found in many of his publications, such as Rieppel, 1994, 1999, and 2000. Considering his authority and influence in our knowledge of Triassic sauropterygians, and for better comparisons with some old literature, we adopted this definition in our manuscript. Yes, we know that normally a “crown group” must be identified by modern taxa, and correspondingly, a “stem group” refers to the paraphyletic group basal to the crown members within the same clade. In our last reply, we would like to explain why we followed Rieppel's definition, when some other papers also used this definition referring to stem/crown group sauropterygians (e.g., Jiang et al., 2019; Berrocal-Casero et al., 2023). However, we understand your criticism of this unusual definition, because it is only used in some studies of Triassic sauropterygians rather than other groups, including the Jurassic and Cretaceous sauropterygians (plesiosaurs). We accept your suggestion here, and have corrected this “wrong” terms and rephrased those related sentences in our manuscript (please see L22-L25; changed “stem” to “basal” in L234, L441), and would avoid to use it anymore in our future studies.

3. Phylogenetic placement and tests of robusticity.

This is one of the most problematic points in my opinion. The new taxon described, Lijiangosaurus, resembles, in its general body proportions, to a pistosauroid. Moreover, in the phylogenetic analyses, it clusters with Wangosaurus, a problematic taxon which the authors regard as either a nothosaur or a pistosauroid. This is confusing because, at the same time, the authors say that their phylogenetic results securely place Lijiangosaurus as a nothosaur.

Because most of importance of the paper relies on the fact that Lijiangosaurus is a long-necked nothosaur, the authors should do everything in their power to dispel the possibility that Lijiangosaurus is a pistosauroid; to see how confident they can be in assessing its phylogenetic position. I suggest two things: test the phylogenetic position of Lijiangosaurus without Wangosaurus in the dataset and make templetton tests to see how many additional steps would be required to have the Lijiangosaurus + Wangosaurus clade within pistosauroids.

A: In fact, we have paid much attention to this issue and your suggestion in the first round of review, following which we have spent time on conducting the phylogenetic analysis using the dataset in Hu et al. (2024) with the result added in our revised version, and have clearly compared the contents and features of multiple matrices focusing on sauropterygians, and also have labeled the values of Bremer support and Bootstrap on each node. For more test, We have run Jackknife test and put the tree with values here.

[figure redacted]

Notably, the result from employing the matrix of Hu et al. (2024), as you suggested, indicates that *Lijiangosaurus* is a nothosaur in the position far from the early-diverging clade of pistosaurs (please see revised Figure S9 and supplementary data). Results from two different latest matrices independently place *Lijiangosaurus* in Nothosauria, which is the evidence stronger than any tests. Please double check the response in our last revision, and please check the synapomorphies of Nothosauria listed in our manuscript that are present in *Lijiangosaurus* but absent in pistosaurs. We would like to advocate that the nothosaurian affiliation of *Lijiangosaurus* is not problematic based on our anatomical comparisons and phylogenetic analyses, and we are confident about that *Lijiangosaurus* is a nothosaur rather than a pistosaur.

As for the position of *Wangosaurus*, we could say that *Wangosaurus* is a nothosaur based on our results independently supported by two different matrices in this study. However, we remain cautious about this statement in this manuscript, because we know that other colleagues are restudying the holotype of *Wangosaurus* with same unpublished conclusion of its nothosaurian affinity and it is out of the scope of our study here. To reduce confusion, we modified the statement of the assumptive pistosaur affinity of *Wangosaurus* in the Discussion (Line 299, 316, 321). We have no reason to exclude the data of *Wangosaurus* in our analyses since it is informative and coded based on our observation on the holotype, and

you can check the skull of *Wangosaurus* in ventral view clearly since we have attached in our last response letter.

If you are still suspicious about the positions of *Lijiangosaurus*, please check the accuracy of our scores in the matrices and test whether you could repeat our result. Considering other reviewers believe the phylogenetic position of *Lijiangosaurus* within Nothosauria, we think they agree the synapomorphies of Nothosauria and could repeat our results by running the dataset provided in the Supplemental Data.

L72-77: modifying clade definitions requires solid tests of topologies, otherwise nearly every paper would have to modify some clade definitions. Could you please indicate which tests you did to assert the robustness of the findings, why the previous definition was problematic, and why it is important to redefined Nothosauria phylogenetically now?

A: Yes, we have modified the clade definition of Nothosauria in our revision, considering one of the other reviewers asked us to do and we agreed that it was worthy to add this part. For the robustness of our result, please see what we replied above. As another reviewer mentioned, "*In the Systematic paleontology section the authors mention 'Nothosauria Baur, 1889', following Rieppel 2000, I assume. Baur (1889) included solely the family name 'Nothosauidae', and the authorship of 'Nothosauria' is dubious (referred to von Zittel or Broili by various works). I advise the authors to properly reformulate the definition of the latter taxon, having in mind their new results. The authors already mention the key synapomorphies of the group, but a proper, well outlined, taxonomic emendation would be useful for future studies*", and considering many newly erected taxa referred to Nothosauria/Nothosauoidea/Nothosauridae and other eosauropterygians reported in recent twenty years, we would better redefine Nothosauria here to clarify which clade we discuss about as "nothosaurs" and for better comparisons in the future studies on related taxa.

4. Assessing convergence.

In the text, convergence is mentioned without any test of it, and in the rebuttal letter, it is said that "the common ancestor of *Lijiangosaurus*, *Wangosaurus*, and pistosaurs developed a neck with no more than 25 cervical vertebrae, which indicates that the clade including *Lijiangosaurus* and *Wangosaurus* evolved elongate necks (more than 30 cervical vertebrae) independently compared to those in pistosaurs"

This is a simplistic interpretation of how convergence works because it boils down to saying that if two taxa have higher counts than an ancestral value, then it is convergent. So would be an elasmosaur and a *Diplodocus*. What if cervical count fluctuates chaotically? You could have two taxa with higher values without them being necessarily, significantly convergent. This is why such tests are important: to see how much it could be due to chance VS a likely real, rapid evolution towards a similar phenotype and likely similar function.

From their rebuttal letter, it seems the authors are not willing to test convergence so we would invite to add more caveats and mention that it is possibly convergent or something along those lines.

A: After discussing with all the authors, we have decided and omitted all the statement of convergence in this manuscript, because it is not a main conclusion in this paper, and our data and methods seem premature to investigate it. We hope you find our current report of this unusual Triassic eosauropterygian valuable even without the a few sentences discussing convergence, and the convergence among these taxa is open for you to test with larger sampling and adequate techniques.

5. Body parts and morphospace occupation.

Firstly, we think it is important to show the full slab (as the authirs did in your rebuttal letter) in the supplementary information.

A: The image showing the full slab of this fossil skeleton has been added in the revised supplementary information.

In their rebuttal letter, the authors are arguing that despite being not preserved, the torso and the tail of the new taxon can be measured. To do so, they drawn a torso that is parallel to the slab edge and measured this length. However, all the trunk and tail bones preserved and the orientation of the caudal centra indicate it was not the case (i.e. that the slab did not break exactly next to the torso). Therefore, it is actually impossible to estimate reliably the length of the torso or of the tail. The authors argue that it is difficult but they did the best they could do. We understand that this is difficult, but the question remains: why create a morphospace with ambiguous values that are impossible to know with sufficient precision?

A: We add some photos here to show the slab from side views. Several neural spines of the dorsal vertebrae are preserved near the margin of the limestone block, and the trunk seems curved along the margin. Nevertheless, we admit that the accurate length of each region is impossible to obtain and any estimation is more or less subjective. Considering the comments from another reviewers querying our PCA results too, therefore, we agree that it is better not to do the analysis with the measurement here. The figures, discussion, and method related to PCA have been deleted in our manuscript and supplementary information, which has no affect on our conclusions..

[image redacted]

We pointed out an important methodological error during the first round of reviews regarding the use of absolute measurements to generate a morphospace. The authors acknowledged it, but, as for many other comments we did, discarded this comment, saying here that they will do a better analysis in a future study. We are sorry but as reviewers, we cannot let the authors use inappropriate methods and erroneous interpretations because they do not master the required methods/basics of multivariate analyses. If the data and the methods are not sound, then it is best not to use them in this paper. If size is indeed the main factor driving PC1, then your morphospace showing PC2+PC3 explains 11% of the total variance, which is extremely small.

As a matter of fact, we did a new analysis using corrected, relative values. First, we accessed the supplementary information of Wang et al 2022 to understand the meaning of the abbreviations. Then, we created a novel dataset using the spreadsheet provided as supplementary information, by dividing all absolute measurements by trunk length. We then z-scaled the dataset and ran a PCA, as well as a distance-based PcoA to generate a morphospace. The morphospace (and its possible interpretation) differs from the morphospaces presented in the figure 5 (see below the PCoA we obtained).

A: Thank you for practicing this new PCA to show some differences from our result, while the *Lijiangosaurus* and *Wangosaurus* are also close to *Yunguisaurus* and *Plesiosaurus* similar to our previous result. Even though we deleted the PCA based on the measurement in this paper, we would like to give some comments on the “methodological error” that you mentioned about why we didn’t use relative values. We agree that it is normal to do PCA with ratios or relative values, but we find it is difficult, if not impossible, to choose a part to standardize the measurements of other parts to reduce the impact from the specimen size. Here you use the trunk length to divide all the other absolute measurements, but we notice that the trunk

length hardly reflects the specimen size, when the proportion of trunk length obviously varies in different Triassic sauropterygians (for example, over 50% in the placodont *Henodus* and around 20% in the plesiosaur *Yunguisaurus*).

Other minor comments in the introduction:

L21: “dominated”: do you mean they are at the top of the food chains? If so, it was possibly the case for a shorter period than 180 million years.

A: We changed “dominated” to be “existed in”.

L30: what, in the modern understanding of evolutionary processes, is an intermediate evolutionary form? The references cited are old general books about reptilian evolution.

A: This description is invalid in modern cladistics, and we have deleted this statement. This sentence is modified to be “and they are generally larger than pachypleurosaurs but smaller than plesiosaurs, including plesiosaurs, in terms of body size.”

L32: “adaptation” instead of “adaption”.

A: It has been corrected/deleted.

Response Letter

Reviewer #4 (Remarks to the Author):

Q: Lines 243-246. “30 cervical vertebrae, contrary to what was previously suggested, is no longer a synapomorphy for pistosaurs or plesiosaurs after our discovery of Lijiangosaurus”. This statement is untenable based on your tree topology and ancestral states analysis, both of which imply an independent homoplastic acquisition of increased cervical number in Lijiangosaurus, but retain pistosaurs+plesiosaurs as a discrete monophyletic clade that presumably still shares increased cervical number as a synapomorphy. You must include the apomorphy lists for at least this key node in the tree to demonstrate whether there has actually been a character suite change involving loss of this shared feature.

A: Based on our phylogenetic analysis, we have attached the synapomorphies mapped on each node (please note that character numbering begins at 0 due to the default setting of the TNT program; these numbers should be incremented by 1 to match the numbering in our character list). As detailed in the Systematic Paleontology section (lines 77-81), we have summarized the synapomorphies (Characters 18, 35, 58, 87, and 115) for the Nothosauria clade. Given that this study primarily focuses on nothosaurs rather than pistosaurs, we have not included the synapomorphies for the Pistosauroidea clade.

We maintain that the statement in lines 244-247 is supported by our phylogenetic results. Our analysis shows that neither the Pistosauroidea clade (pistosaurs+plesiosaurs) nor the node leading to plesiosaurs includes the transition to more than 30 cervical vertebrae (Character 98: 0→1) as a synapomorphy. This observation is particularly relevant as Wintrich et al. (2017, Figure 3B) previously identified an increased cervical count (>30 vertebrae) as a distinguishing feature of Pistosauroidea (including Plesiosauria) compared to other eosauropterygians.

We believe this statement remains valid based on our findings, but we would be pleased to provide additional clarification if the reviewer has further concern regarding this interpretation.

[image redacted]

Q: Lines 256–258: This entire paragraph is awkwardly written and repeats information from earlier in the text. I suggest deleting it and beginning the section with “in the ventral view of the skull” from line 269.

A: We appreciate this helpful suggestion. We agree that this paragraph was redundant as it repeated previous content, and we have therefore removed it. The section now begins directly with the subsequent paragraph (line 256).

Q: Lines 314–315: You might also wish to note that other recent Bayesian phylogenies have resolved *Wangosaurus* as a nothosaurid (see Kear et al. 2024 *Curr Biol* 34, R553 – R563).

A: We agree this is indeed an important recent publication relevant to our study. Not only does it support our identification of *Wangosaurus* as a nothosauroid, but it also provides the report of a zygantrum (an accessory intervertebral articulation) in a nothosaur vertebra. As this work was published after our initial submission and previous revisions, we have now appropriately incorporated citations to it in our manuscript (lines 300 and 389).

Q: Lines 372–374: This sentence integrates conflicting statements. *Lijiangosaurus* does not record “the first independent origin of the remarkably long neck” “in the whole sauropterygian evolutionary history” since increasing cervical numbers demonstrably occur in different plesiosauroid clades (e.g., cryproclidids, microcleidids, and elasmosaurids). This seems to be acknowledged in the second part of the sentence with “being earlier than the arising of pistosaurs and descendant plesiosaurs”. What are the authors trying to say? Please clarify and emend accordingly.

A: The development of elongated necks with over 30 cervical vertebrae is well-documented in these plesiosauroid clades, while all such occurrences are in post-Triassic taxa that postdate *Lijiangosaurus*. We acknowledge that our original phrasing may have been unclear, and have accordingly revised the sentence (lines 353–356) to be:

“within sauropterygian evolution history, *Lijiangosaurus* represents the earliest occurrence of an exceptionally long neck (comprising over 30 cervical vertebrae), predating the emergence of both basal pistosaurs and their plesiosaur descendants”.

Q: Lines 374–377: “Since the function of the long neck in plesiosaurs and their pistosaur ancestors is undetermined, the long neck reported here provides evidence indirectly supporting that the neck elongation in nothosaurs and pistosaurs seems beneficial for foraging behavior”. How? This statement is

ambiguous and needs to be explained in more detail.

A: We have modified this sentence (lines 356–361) to be:

“given that the function of elongated necks in plesiosaurs and their plesiosaur ancestors remains uncertain, the presence of an elongated neck in *Lijiangosaurus*, which is neither a fast swimmer nor a pursuit predator, provides indirectly support for the foraging benefit hypothesis of neck elongation in sauropterygians, facilitating ambush predation and larger feeding ranges.”

To clarify this discussion, we have emphasized that a markedly elongated neck evolved in *Lijiangosaurus*, a slow-swimming and potentially semi-aquatic sauropterygian, which aligns well with the foraging function hypothesis. Additionally, we have incorporated one more reference about this hypothesis of long necks in plesiosaurs.

–Wilkinson, D. M. & Ruxton, G. D. Understanding selection for long necks in different taxa. *Biol Rev*, 87(3), 616–630 (2012).

Reviewer #5 (Remarks to the Author):

Q: The authors have dealt with the various minor and major criticisms raised by the Reviewers, especially Reviewer 1, to make sure statements in the text are accurate and clear. I have no further comments.

A: Many thanks.

Reviewer #2 (Remarks to the Author – feedback from the previous round of reviews):

The main points of my review have been taken into account by the Authors. My minor comments included in the submitted manuscript have also been addressed. Overall, I am satisfied with the Authors’ approach to my (hopefully constructive) criticism. I think that the manuscript has benefited from the reviews, and I appreciate that the Authors took their time to perform the ancestral state reconstruction and another phylogenetic analysis. Some minor issues persist:

Q: Please read through the Supplementary Manuscript again and correct all the minor mistakes that are still contained within it (e.g. third sentence “Although the posterior dorsal, the sacral, and the proximal and distal [CAUDAL, I assume] vertebrae are missing possibly due to weathering [...]”; the explanation of ‘cqp’ marked on the Figure S3 is missing etc.);

A: Many thanks for your careful review. The word ‘caudal’ has been added in the third sentence as mentioned. Figure S3 has been changed to be Figure S4 because of additional figure inserted in our revision, and the abbreviation of ‘cqp’ as ‘cranio-quadrato passage opening’ has been added in the caption.

We have gone through the Supplementary Information, and have revised the figure numbers accordingly in the text, rephrased some sentence with several words

corrected (e.g., line 34-35, line 72, line 81-82, line 109-110, line 148-149, etc.), and deleted some redundant words (e.g., line 48 ‘natural’ , line 90 ‘reptiles’ , etc.) based on suggestions from a native English speaker.

Q: Lines 42-43 in the main text: “[...] the only exceptions convergently resembling plesiosaurs but with completely different cervical shapes are a few bizarre early-diverging archosauromorphs [...].” Once more, this is not true. Archosauromorphs are not the ONLY exceptions, due to *Hyphalosaurus* not being an archosauromorph. Please reformulate (e.g. “Some reptile lineages (e.g. tanysaurians) have achieved a general bauplan similar to that of plesiosaurs, but with completely different vertebral morphology (citation).”);

A: Yes, *Hyphalosaurus* has similar long-necked bauplan and it is not an archosauromorph, while it is not a marine reptile. To be precise, we rewrote this sentence as (lines 41-44):

“Certain early-diverging archosauromorph taxa (e.g., *Dinocephalosaurus*, *Tanystropheus*) convergently achieved general body plans resembling those of long-necked plesiosaurs, while these archosauromorphs evolved fundamentally distinct vertebral morphologies (Spiekman et al., 2024)” .

Q: Line 75 *Keichousaurus* instead of “*Kechousaurus*”;

A: The spelling has been corrected (line 70).

Q: Lastly but importantly, I was still not able to reproduce the results of the PCA. After log-transforming the supplied data, and performing both correlation or variance-covariance based PCA in Past 4.15, the PC variance yields differ, in both cases, from those provided in the manuscript. I assume that some miniscule changes have been made to the supplementary table after the PCA had been carried out. These are very minor differences (1% point), but nevertheless this issue should be addressed by the Authors to improve the quality of their work, especially if they insist on keeping the PCA within the main text, despite the concerns shared by me and at least one of the other Reviewers.

A: According to comments from Reviewer #1, we have omitted these analysis and result of PCA, when it has little influence of the key conclusions of this study.

To reiterate, I think that the manuscript’s quality has significantly improved and the remaining issues can be easily solved. I congratulate the Authors on this important publication and hope to see it published soon!

A: Thank you again for your constructive review and positive words.

Dear editor, dear authors,

We have reviewed this new version of the paper. Sadly, we see in the revised version as well as in the rebuttal letter that the authors did not wish to substantially modify the paper despite important points and errors we discussed during the first round of reviews.

We will detail the main points below, but as the authors do not wish to modify the paper and maintain erroneous terminologies and erroneous analytical protocols, we will again recommend rejection. We do not want to review it again unless it is substantially modified.

1. Implications for plesiosaurian evolution

In the main text and in the rebuttal letter, it is said “*we hope to have implications on our understanding of the early evolution of Triassic eosauropterygians, as well as their descendant plesiosaurs.*”

This is an enduring misconception present in the paper: how can a peculiar morphology in a single nothosauroid have implications for the evolution of plesiosaurians (which do not descend from nothosaurs)? We understand the implications for eosauropterygia as a whole, that it is indeed very interesting to have an earlier example of neck elongation in eosauropterygians. Perhaps that is what you mean but then we would suggest to clearly state it in the abstract, intro, discussion, and conclusion.

L5, L48: it is said in the abstract and the introduction that the origin of the long neck in plesiosaurians is controversial. It is not true, it arose thanks to a combination of somitogenesis and differential growth and was already present, to a lesser extent, in early pistosauroids.

L51: once again, the authors are seemingly suggesting that things that are actually known, aren't. If the “hydrofoil bodyplan” they mention it the bodyplan with four long flippers, then it evolved concomitantly than a fairly long neck in pistosauroids.

2. The use of incorrect phylogenetic terms.

In the rebuttal letter, the authors advocate for the use of “stem” in Sauropterygia and indicate that a crown group exist as well (L23-25). This is a blatant error. A crown group can only be defined by extant taxa, and the lineages basal to the crown is the paraphyletic stem group. We really cannot understand why the authors are willing to use an erroneous terminology because it was mentioned in a paper 24 years ago.

3. Phylogenetic placement and tests of robusticity.

This is one of the most problematic points in my opinion. The new taxon described, *Lijiangosaurus*, resembles, in its general body proportions, to a pistosauroid. Moreover, in the phylogenetic analyses, it clusters with *Wangosaurus*, a problematic

taxon which the authors regard as either a nothosaur or a pistosauroid. This is confusing because, at the same time, the authors say that their phylogenetic results securely place *Lijiangosaurus* as a nothosaur.

Because most of importance of the paper relies on the fact that *Lijiangosaurus* is a long-necked nothosaur, the authors should do everything in their power to dispel the possibility that *Lijiangosaurus* is a pistosauroid; to see how confident they can be in assessing its phylogenetic position. I suggest two things: test the phylogenetic position of *Lijiangosaurus* without *Wangosaurus* in the dataset and make templetton tests to see how many additional steps would be required to have the *Lijiangosaurus* + *Wangosaurus* clade within pistosauroids.

L72-77: modifying clade definitions requires solid tests of topologies, otherwise nearly every paper would have to modify some clade definitions. Could you please indicate which tests you did to assert the robustness of the findings, why the previous definition was problematic, and why it is important to redefined Nothosauria phylogenetically now?

4. Assessing convergence.

In the text, convergence is mentioned without any test of it, and in the rebuttal letter, it is said that "*the common ancestor of Lijiangosaurus, Wangosaurus, and pistosaurs developed a neck with no more than 25 cervical vertebrae, which indicates that the clade including Lijiangosaurus and Wangosaurus evolved elongate necks (more than 30 cervical vertebrae) independently compared to those in pistosaurs*"

This is a simplistic interpretation of how convergence works because it boils down to saying that if two taxa have higher counts than an ancestral value, then it is convergent. So would be an elasmosaur and a *Diplodocus*. What if cervical count fluctuates chaotically? You could have two taxa with higher values without them being necessarily, significantly convergent. This is why such tests are important: to see how much it could be due to chance VS a likely real, rapid evolution towards a similar phenotype and likely similar function.

From their rebuttal letter, it seems the authors are not willing to test convergence so we would invite to add more caveats and mention that it is possibly convergent or something along those lines.

5. Body parts and morphospace occupation.

Firstly, we think it is important to show the full slab (as the authors did in your rebuttal letter) in the supplementary information.

In their rebuttal letter, the authors are arguing that despite being not preserved, the torso and the tail of the new taxon can be measured. To do so, they drawn a torso that is parallel to the slab edge and measured this length. However, all the trunk and tail bones preserved and the orientation of the caudal centra indicate it was not the case (i.e. that the slab did not break exactly next to the torso). Therefore, it is actually impossible to estimate reliably the length of the torso or of the tail.

The authors argue that it is difficult but they did the best they could do. We understand that this is difficult, but the question remains: why create a morphospace with ambiguous values that are impossible to know with sufficient precision?

We pointed out an important methodological error during the first round of reviews regarding the use of absolute measurements to generate a morphospace. The authors acknowledged it, but, as for many other comments we did, discarded this comment, saying here that they will do a better analysis in a future study. We are sorry but as reviewers, we cannot let the authors use inappropriate methods and erroneous interpretations because they do not master the required methods/basics of multivariate analyses. If the data and the methods are not sound, then it is best not to use them in this paper. If size is indeed the main factor driving PC1, then your morphospace showing PC2+PC3 explains 11% of the total variance, which is extremely small.

As a matter of fact, we did a new analysis using corrected, relative values. First, we accessed the supplementary information of Wang et al 2022 to understand the meaning of the abbreviations. Then, we created a novel dataset using the spreadsheet provided as supplementary information, by dividing all absolute measurements by trunk length. We then z-scaled the dataset and ran a PCA, as well as a distance-based PcoA to generate a morphospace. The morphospace (and its possible interpretation) differs from the morphospaces presented in the figure 5 (see below the PCoA we obtained).

Other minor comments in the introduction:

L21: “dominated”: do you mean they are at the top of the food chains? If so, it was possibly the case for a shorter period than 180 million years.

L30: what, in the modern understanding of evolutionary processes, is an intermediate evolutionary form? The references cited are old general books about reptilian evolution.

L32: “adaptation” instead of “adaption”.

All the best,

Valentin Fischer
Antoine Laboury